# Multiscale Analysis of Sandwich Beams with Polyurethane Foam Core: A Comparative Study of Finite Element Methods and Radial Point Interpolation Method

**DOI:** 10.3390/ma17184466

**Published:** 2024-09-11

**Authors:** Jorge Belinha

**Affiliations:** 1Department of Mechanical Engineering, ISEP, Polytechnic of Porto, Rua Dr. António Bernardino de Almeida, n. 431, 4200-072 Porto, Portugal; job@isep.ipp.pt; 2Institute of Science and Innovation in Mechanical and Industrial Engineering (INEGI), Rua Dr. Roberto Frias, 4200-465 Porto, Portugal

**Keywords:** meshless methods, radial point interpolation method, homogenisation, multiscale, sandwich structures

## Abstract

This study presents a comprehensive multiscale analysis of sandwich beams with a polyurethane foam (PUF) core, delivering a numerical comparison between finite element methods (FEMs) and a meshless method: the radial point interpolation method (RPIM). This work aims to combine RPIM with homogenisation techniques for multiscale analysis, being divided in two phases. In the first phase, bulk PUF material was modified by incorporating circular holes to create PUFs with varying volume fractions. Then, using a homogenisation technique coupled with FEM and four versions of RPIM, the homogenised mechanical properties of distinct PUF with different volume fractions were determined. It was observed that RPIM formulations, with higher-order integration schemes, are capable of approximating the solution and field smoothness of high-order FEM formulations. However, seeking a comparable field smoothness represents prohibitive computational costs for RPIM formulations. In a second phase, the obtained homogenised mechanical properties were applied to large-scale sandwich beam problems with homogeneous and approximately functionally graded cores, showing RPIM’s capability to closely approximate FEM results. The analysis of stress distributions along the thickness of the beam highlighted RPIM’s tendency to yield lower stress values near domain edges, albeit with convergence towards agreement among different formulations. It was found that RPIM formulations with lower nodal connectivity are very efficient, balancing computational cost and accuracy. Overall, this study shows RPIM’s viability as an alternative to FEM for addressing practical elasticity applications.

## 1. Introduction

In modern engineering, sandwich structures are extensively applied to construct lightweight laminated structures allowing for advanced mechanical solutions in naval, automotive, aerospace, and aeronautic applications [1]. Among structural laminated solutions, sandwich structures stand out as a special type of laminate, which are characterised by two thin high-rigid face sheets (e.g., fibre-reinforced composite or aluminium) enclosing a soft thick core. Generally, its core is manufactured with low-stiffness and low-density materials. Nevertheless, being confined by two rigid materials allows the core to increase the bending rigidity of the structural set while keeping the overall weight of the sandwich structure low. Several factors influence the structural mechanical behaviour of sandwich structures, such as the microstructure of the core material; the thickness ratio of core and face sheets; the fibre volume ratio; and the material (and its mechanical orientation) of the face sheets [1,2]. Due to its mechanical complexity, the development of numerical methods capable of efficiently and accurately predicting the structural behaviour of sandwich structures is a never-ending task.

In this work, due to its versatility, polyurethane foam (PUF) has been selected as the core material. It is possible to find in the literature several research studies on PUF [3], which is a widely used material in the polymer family due to its exceptional properties, such as lightweight, eco-friendliness, low density, shock absorption, processability, and elasticity. These foams have a cellular structure with polyhedral cells that fill three-dimensional space, which, combined with various manufacturing techniques, make it possible to design PUF with specific mechanical properties to be applied to particular applications [4]. It is possible to produce PUF in multiple physical states and densities to meet specific needs, making it cost-effective for diverse uses. PUFs are commonly used as core materials in sandwich structures integrating aerospace, aeronautic, naval, automotive, and construction structures and mechanisms [3].

Most commonly, sandwich panels and laminates are analysed using equivalent single-layer (ESL) theories [5], in which the laminate thickness is modelled using a transverse deformation theory. Thus, the problem domain can be represented with a 2D domain. This dimensional simplification makes it possible to reduce the computational cost of the analysis when compared with a full 3D deformation elasticity theory for 3D discretisations. However, 3D solutions allow for more realistic physical simulations and predictions than 2D ESL theories, particularly for thick plates and shells [6].

Thus, several analytical solutions using 3D elasticity theory have been developed to investigate the bending response of sandwich plates. In the pioneer research work of Pagano [7], exact 3D solutions were derived for the stress analysis of simply supported rectangular sandwich plates. Later, Zenkour [8] applied 3D elasticity equations to obtain the bending analytical solutions of rectangular multilayer plates under transversal distributed loading conditions. For sandwich plates with a functionally graded core, Kashtalyan and Menshykova [9] proposed a 3D exact elasticity solution to predict their bending behaviour for sinusoidally distributed transverse loads. Woodward and Kashtalyan further developed the 3D exact elasticity solution to predict the bending response of sandwich plates under localised transverse loads [10] and many other relevant transversal loading scenarios [11].

Discretising with full detail the cellular material forming the sandwich core will lead to numerical analyses with high computational costs. Thus, multiscale homogenisation techniques are generally applied to reduce the time of the structural analyses, making it possible to obtain fair approximated solutions [12]. Homogenisation techniques provide a multiscale analysis by assuming the existence of multiple spatial scales in materials and structures. Commonly, the analysis of heterogeneous materials has relied on effective properties derived from homogenising the response at microscopic scales, which are then transported to a macroscale analysis. Within the framework of small deformations and linear elasticity, numerous analytical models have been developed to predict the homogenised constitutive response of heterogeneous materials at the macroscopic level, incorporating the characteristics of their microstructure [12]. These analytical models are based on the Hill–Mandel condition for homogeneity [13]. This principle states that, when dealing with materials with vastly different microscopic and macroscopic length scales, the volume-averaged strain energy within a representative volume element (RVE) can be computed as the product of the volume-averaged stress and strain fields within that same RVE. The Hill–Mandel condition demonstrates the energy equivalency between the homogeneous and the heterogeneous material [13,14]. Micromechanical analysis of two-phase materials often utilises discretisation-based approaches to predict the overall response from their microstructure [15].

In Computational Mechanics, the finite element method (FEM) is the most popular discretisation technique. Since its origin, the FEM has been successfully applied to several engineering fields [16]. Nevertheless, the computational mechanics research community employs continuous efforts to discover and develop new discretisation techniques possessing higher efficiency and accuracy. Today, meshless methods (or meshfree methods) are alternative discretisation techniques, which are capable of replacing the FEM in most applications [17,18]. In meshless methods, the solid domain is discretised with an unstructured nodal set instead of the standard element mesh of FEM [19,20]. With the FEM, the nodal connectivity is obtained by means of the element concept. Differently, in meshless methods, such connectivity is achieved using the influence domain concept [21].

The diffuse element method (DEM), proposed by Nayroles et al. as an generalisation of the FEM, was the first developed mature meshless method (or meshfree method) [22] using moving least square (MLS) approximants. Later, Belytschko et al. slightly enhanced the DEM and extended it to elasticity problems [23]. By introducing a background integration mesh based on the Gauss–Legendre quadrature scheme and applying the Lagrange multipliers to impose the boundary conditions, Belytschko et al. called their DEM version the Element-Free Galerkin Method (EFGM) [23], becoming one of the most popular meshless method ever developed. After, in the same decade, other meshless methods appeared, such as the Reproducing Kernel Particle Method (RKPM) [24] or the Meshless Local Petrov–Galerkin Method [25]. By using approximant shape functions and higher nodal connectivity, these meshless methods are capable of delivering smoother variable fields and more accurate solutions [21]. Although all these meshless methods were successfully applied to solid and fluid mechanics, their approximant shape functions lack the delta Kronecker property, hindering the direct imposition of natural and essential boundary conditions. In these approximant meshless methods, the most usual technique to impose the boundary conditions involes applying the Lagrange multipliers, which increases the size of the system of equations and, consequently, the overall cost of the analysis [21]. Compared to the FEM, this is a significant disadvantage.

Thus, the research community started to focus their efforts in the development of meshless methods with interpolating shape functions. Proposed by Sukumar et al., the Natural Element Method (NEM) was one of the first interpolant meshless methods [26] and a truly meshless method. With the NEM, the natural and essential boundary conditions could be exactly imposed with the same techniques applied to the FEM, reducing the overall cost of the meshless analysis. Afterwards, other ingenious interpolant meshless methods were developed, such as the Point Interpolation Method (PIM) [27], the Radial Point Interpolation Method (RPIM) [28], the Meshless Finite Element Method (MFEM) [29], and the Natural Radial Element Method (NREM) [30]. In the field of interpolant meshless methods, the RPIM is certainly the most popular technique [21].

Later, combining the connectivity technique of the NEM with the interpolation functions of the RPIM, Dinis et al. developed a truly meshless method: the Natural Neighbour Radial Point Interpolation Method (NNRPIM) [31]. The NNRPIM only requires the nodal discretisation of the problem’s domain. Afterwards, using that spatial information, it autonomously distributes the integration points and establishes nodal connectivity. There is no need for an independent background integration mesh, as in the EFGM or RPIM.

In the literature, it is possible to find several research works addressing the combination of meshless methods with ESL deformation theories for the analysis of sandwich structures [32]. However, meshless formulations using a 3D deformation theory are not so common [6,33,34]. Regarding multiscale homogenisation techniques combined with meshless methods, Rodrigues et al. extended the RPIM [35] and the NNRPIM [36] to the multiscale analysis of laminated composites, and Wang et al. developed a multiscale approach for the computational modelling of the mechanical behaviour of carbon nanotube-reinforced cement composites [37].

Regarding the main novelty of this study, for the first time, this work combines the RPIM with a multiscale homogenisation technique to predict the macroscale behaviour of sandwich structures. This research aims to compare the performance and the efficiency of the RPIM with the FEM (computational cost, accuracy, stress field distribution) and to explore the meshless multiscale analysis of sandwich structures with a homogeneous or a functionally graded foam core. Thus, firstly, the homogenised mechanical properties of the cellular units are estimated and correlated with the foam density. Then, at a macroscale, distinct sandwich structures are modelled by varying the density distributions along the sandwich thickness. The obtained results are compared with FEM solutions. This work is a preliminary work on the structural analysis of sandwich structures using the RPIM, which is intended to understand if the RPIM is sufficiently accurate and robust in linear elastostatic applications of sandwich structures. If the accuracy and robustness of the RPIM is verified, it can be extended to more demanding applications, such as material and geometrical nonlinearities. Beyond the elastic limit, sandwich structures start to show elastoplastic effects (and damage) in the core and a consequent crushing. In computational analysis, such nonlinear behaviour will produce severe mesh distortions, which the FEM cannot handle efficiently. In opposition, meshless methods are capable of dealing with mesh distortions without requiring any remeshing.

This document is divided in four sections. In Section 1, the state-of-the-art of sandwich structures are presented and their corresponding mathematical formulation, multiscale homogenisation approaches, and usual dicretization techniques. In Section 2, the meshless method mathematical formulation for elasticity is presented, as well as the adopted homogenisation technique. Then, in Section 3, the numerical results are presented and discussed. First, the obtained cellular foam homogenised mechanical properties with respect to their apparent density are presented. Afterwards, macroscale benchmark examples of sandwich structures are presented and their solutions discussed. In the end, the main conclusions and remarks are presented in Section 4.

## 2. Numerical Method

### 2.1. Meshless Methods

Meshless methods employ a well-established discretisation approach distinct from mesh-based methods [20,21]. Unlike predefined meshes with element connectivity, for domain discretisation, meshless methods rely on a set of nodes, which can be distributed either regularly or irregularly. The accuracy of the meshless solution hinges on the nodal density, which is similar to how mesh density impacts accuracy in mesh-dependent methods using weak formulations for elasticity [21]. Generally, as the nodal density increases, the corresponding solution becomes more accurate. Furthermore, for problems with specific geometric features or concentrated loads that significantly influence stress distribution, a refined nodal density in those regions becomes crucial to improve the meshless method’s accuracy. Following the nodal discretisation, a set of background integration points is defined across the domain, which is known as the integration mesh.

Unlike finite element methods (FEMs) that rely on predefined element connectivity, meshless methods require an alternative approach to establish relationships between nodes and integration points. This is achieved through the concept of the influence domain [21]. In the influence domain approach, each integration point, denoted by $x_{I}$, identifies a set of nodes within a predefined radius. These nodes will be used to construct the interpolation function of its corresponding integration point, $x_{I}$, forming its influence domain. This concept offers a simple technique to establish the nodal connectivity in meshless methods, and it is employed by various meshless formulations (e.g., the RPIM, EFGM, MLPG, and RKPM). The two main approaches for defining influence domains are fixed or variable radial search. Within a fixed radial search, a constant radius for all integration points is used. While simple to implement, it can lead to inconsistencies near boundaries. Integration points close to physical boundaries may have fewer nodes within their influence domain compared to those located within the domain, which can affect the accuracy of the solution. Alternatively, a variable radial search aims to ensure a consistent number of nodes influencing each integration point. This is achieved by adjusting for each integration point $x_{I}$ the searching radius, which will make it possible to maintain the same number of nodes within the influence domain of all integration points. Although this technique enforces that all integration points will construct shape functions with the same order, it can be computationally more expensive due to the dynamic radius calculation.

After establishing the nodal connectivity of each integration point, shape functions functions are constructed, making it possible to approximate/interpolate the variable field at an integration point $x_{I}$ with
(1)$$u\left(x_{I}\right)=\sum_{i=1}^{n}\varphi_{i}\left(x_{I}\right)\cdot u\left(x_{i}\right)=\Phi {\left(x_{I}\right)}^{T}\cdot u_{s}=\left\{\varphi_{1}\left(x_{I}\right),\varphi_{2}\left(x_{I}\right),\cdots ,\varphi_{n}\left(x_{I}\right)\right\}\cdot \left\{\begin{array}{c} u(x_{1})\\ u(x_{2})\\ \cdots \\ u(x_{n}) \end{array}\right\}$$
where the number of nodes within the influence domain of the integration point $x_{I}$ is defined as *n*, the field variable components of each node $x_{i}$ inside the influence domain of $x_{I}$ are contained in $u_{s}$, and $\varphi_{i}\left(x_{I}\right)$ is the *i*th component of the shape function constructed for $x_{I}$. These functions, and their partial derivatives, are then incorporated into the chosen governing equation (strong or weak form) to establish a system of algebraic equations. Solving this system yields the solution variable fields (pressure, displacement, velocity, etc.) across the domain. Details regarding the RPIM (nodal connectivity, background integration mesh, shape functions, and discretisation) and the elastostatic formulations can be found in Appendix A.

### 2.2. Material Homogenisation Technique

The representative volume element (RVE) concept, introduced by Hill [13], offers an efficient tool for the multiscale analysis of materials. An RVE represents a statistically relevant subvolume of the material, capturing the essential microstructural features that govern its overall behaviour. A key aspect is ensuring statistical representativeness: the RVE must encompass a sufficient sampling of all the microstructural features within the material, even for materials with varying constituent volume fractions. The appropriate RVE size remains a critical parameter. It must be large enough to capture the relevant heterogeneities but remain significantly smaller than the macroscopic domain of interest. Appendix B introduces the RVE concept to present a micromechanical approach for determining the effective elastic properties of materials. This approach makes it possible to bridge scales by numerically connecting the microscopic behaviour of the heterogeneous constituents and the homogenised effective behaviour observed at the macroscopic level.

## 3. Numerical Results

In this section, microscale and macroscale examples are analysed, and the obtained results are discussed. First, the homogenised mechanical properties of a polyurethane foam (PUF) sample with circular voids were obtained using the homogenisation procedure described in Section 2.2. Then, using the homogenised mechanical properties, a macroscale application was analysed: a sandwich cantilever beam with aluminium face sheet layers and PUF core. Several PUF densities were analysed, making it possible to observe the effects of considering a constant density for the PUF core or a functionally graded density.

In order to compare the obtained RPIM results, finite element formulations were applied [38]. Thus, in this work, the following FEM formulations were considered:FEM-3n: constant strain linear triangular elements (three-node elements), with one integration point per element.FEM-6n: quadratic triangular elements (six-node elements), with full integration.FEM-4n: Lagrangian four-node elements, with full integration.

Regarding the RPIM formulations, in this work, four RPIM formulations were studied:CRPIM: a classic RPIM considering 16 nodes inside the influence domain, shape parameters c=1.42 and p=1.03, and a linear polynomial basis.MRPIM16: a modified version of the RPIM considering 16 nodes inside the influence domain, shape parameters c=0.0001 and p=0.9999, and a constant polynomial basis.MRPIM9: a modified version of the RPIM considering nine nodes inside the influence domain, shape parameters c=0.0001 and p=0.9999, and a constant polynomial basis.MRPIM4: a modified version of the RPIM considering four nodes inside the influence domain, shape parameters c=0.0001 and p=0.9999, and a constant polynomial basis.

Regardless the RPIM formulation, the following integration schemes were assumed:If the background integration mesh is made with triangular cells, one integration point is considered for each triangular integration cell, located at the triangular cell centre, and with an integration weight $\omega_{I}$ equal to its volume ($\omega_{I}=A_{I}\cdot h_{I}$, with $A_{I}$ being the triangle area and $h_{I}$ being the element thickness).If quadrilaterals are used to build the background integration mesh, the Gauss–Legendre quadrature scheme is adopted, assuming 2×2 integration points per quadrilateral cell (as shown in Figure A1).

All the finite element and meshless methods codes were completely written by the author using Matlab R2017b^TM^. The nodal and element meshes were built with Simcenter Femap Student Edition Software^TM^. All the analyses were performed with a standard laptop computer, with an Intel^©^ Core^TM^ i5-3230M CPU (Samsung, Suwon, Republic of Korea), 2.60 GHz, and 8 GB of RAM.

### 3.1. Microscale Analysis

In this section, the homogenised mechanical properties of PUF with microscale cylindrical holes are investigated. Thus, assuming the plane strain deformation theory and a unitary thickness, a 2D parametric RVE was considered, Figure 1, with L=D=10 mm, $x_{c}=\left\{L/2,D/2\right\}$, and radius *R* of the central hole defined as
(2)$$R\left(v_{f}\right)=\sqrt{\frac{1-v_{f}}{\pi }}$$
in which $v_{f}$ represents the volume fraction of the RVE,
(3)$$v_{f}=\frac{v_{m}}{v_{d}}$$
with $v_{m}$ being the volume occupied by effective PUF mass and $v_{d}$ being the volume of the RVE: $v_{d}=L\cdot D\cdot 1$ mm^3^.

For the bulk PUF ($v_{f}=1$), the isotropic elastic mechanical properties documented in the literature [39] were considered: E=171.43 MPa, G=57.810 MPa, and ν=0.30.

#### 3.1.1. Convergence Study

Since this work proposes a discrete meshless method for a multiscale analysis, it is necessary to verify the convergence of the meshless technique at the microscale. Therefore, a volume fraction of $v_{f}=0.75$ was considered, corresponding to a radius R=2.821 mm. Next, five distinct meshes were constructed, as shown in Figure 2.

For each mesh indicated in Figure 2, the procedure described in Section 2.2 was applied to obtain the elastic components of the homogenised constitutive material matrix, $C_{ij}$, and the homogenised material properties of the RVE: the Young modulus in direction Ox and Oy—$E_{x}$ and $E_{y}$, respectively—and the in-plane elastic shear modulus and Poisson’s ratio—$G_{xy}$ and $\nu_{xy}$, respectively. The obtained results are presented in Figure 3.

Although the final converged value was not the same for the all the techniques, the results of Figure 3 show that all the discretisation techniques converged. Both of the FEM formulations converged to the same value, with FEM-6n being the one showing a higher convergence rate. Similarly, regarding the elastic mechanical property, all the RPIM formulations converged to the same value as well, with the MRPIM4 being the one showing the highest convergence rate and the CRPIM being the formulation with the lower one. It is possible to notice that the FEM formulations presented an upper-bound convergence path, and the RPIM formulations showed a lower-bound path. Some formulations presented very close results. Thus, their convergence curves apparently overlapped. As Figure 3 shows, the CRPIM and MRPIM16 overlapped their solutions in all presented results, and in Figure 3b, the solution of all the RPIM formulations overlapped, being visible in only one line. In order to show how close the mentioned solutions truly are and allow future comparisons, the numerical values of the convergence study are shown Table 1 and Table 2.

Concerning the homogenised elastic mechanical properties obtained for the most refined mesh, in Table 1, it is possible to observe that the difference between the RPIM and FEM formulations is around 3.5% for the Young modulus $E_{x}$ and $E_{y}$, 1.5% for the Young’s modulus $E_{z}$, 6.5% for the elastic shear modulus $G_{xy}$, and 1.2% for the Poisson’s ratio. Since the FEM and RPIM techniques are fundamentally different in their formulations, it is expected to find such differences in the obtained results.

Observing the elastic constitutive constants obtained for the most refined mesh, in Table 2, the output can be seen to follow the same trend. The results of the FEM formulations are very similar between each other, and the RPIMs’ results are all very close to each other as well. Comparing the FEM results with the RPIM solutions, it is possible to observe the following differences: 3.3% for $C_{11}$ and $C_{22}$, 3.0% for $C_{12}$, 1.5% for $C_{33}$, and 6.5% for $C_{44}$.

In order to understand the stress distribution obtained with each formulation, the von Mises equivalent stress field ($\sigma_{ef}=\sqrt{3\cdot J_{2}}$) obtained with each formulation is presented in Figure 4 and Figure 5.

Observing Figure 4 and Figure 5, it is possible to observe that the FEM-6n formulation delivered a very smooth stress distribution. Such a result was expected, because its shape functions possess quadratic terms. Then, the CRPIM and MRPIM16 presented stress distributions very similar with the FEM-3n formulation, which was not expected.

Generally, the literature shows that meshless methods are capable of delivering smoother stress fields [21,23]. However, the obtained RPIM effective stress fields, presented in Figure 4 and Figure 5, apparently contradict such a notion. The stress distributions of the CRPIM and MRPIM16 formulations present the same granulated distribution observed in the FEM-3n formulation. The results became even worst when a lower number of nodes was considered inside the influence domains, as for the MRPIM9 and MRPIM4 formulations (Figure 5d–i).

The reason behind this lack of smoothness comes from the adopted integration scheme. In order to deliver a faster and more flexible RPIM formulation, when triangular integration cells are used, only one integration point is inserted inside the triangular integration cell. Although such a procedure does not significantly affect the accuracy of the variable fields, it affects the smoothness distribution of such fields. When the integration order was increased, using for example four integration points inside each integration cell (following the Gauss–Legendre integration scheme described in [21]), the smoothness of the stress fields significantly increased, as shown in Figure 6.

Comparing the results obtained with the CRPIM and the CRPIM* version with an higher integration order (comparing Figure 4g–i with Figure 5a–c), it is possible to visualise that the stress distribution improved significantly, then becoming more smooth and continuous (the granulated distributions have disappeared).

Nevertheless, although the stress field (and all the other variable fields as well) became smoother, the homogenised values did not change. Table 3 shows the results obtained with the CRPIM* version with an higher integration order. Comparing such results with the solutions presented in Table 1 and Table 2, it is possible to observe that the homogenised values were not affected by the integration scheme.

The main disadvantage of using the higher integration scheme (for any of the RPIM formulations) is the computational cost. Using one integration point inside each triangular integration cell makes it possible to perform analyses about 10 times faster than using four integration points, which becomes relevant when nonlinear analyses are being considered. Thus, the next examples will continue to consider only one integration point inside each triangular integration cell.

Regarding the overall computational cost of the FEM and RPIM formulations, Figure 7 is presented. The displayed data concers the complete computational cost, including the nodal connectivity, construction of the background integration mesh, and construction of the shape functions, thus establishing the global system of equations and obtaining all the variable fields (displacements, strains, and stresses). As Figure 7 shows, the FEM presented a very low computational cost. The fastest RPIM formulation was the MRPIM4. Even so, it was four times slower than the FEM. The CRPIM and MRPIM16 presented the highest computational costs, being almost 10 times slower than the FEM. Nevertheless, it is relevant to clarify that the code written by the author was not written for speed and to achieve computational efficiency. Several programming improvements could be applied to the RPIM code to significantly reduce the overall computational cost. Most of the RPIMs’ computational burden is due to the preprossessing phase (nodal connectivity, construction of the background integration mesh, and construction of the shape functions). In the FEM, the element mesh (containing the information of which node belongs to each element) is already available. In the RPIM, it is necessary to first find the influence domains. Then, in the FEM, the shape functions and their partial derivatives are already available (using isoparametric transformation), so it is not necessary to invert any geometric matrix for each integration point as in the RPIM. Moreover, the FEMs’ shape functions possess three or six nodes, and the RPIMs’ shape functions can possess 16 nodes, which will delay the shape function construction step, as well as the stiffness matrix construction phase. Also, the background integration mesh is already established in the FEM (again using isoparametric transformation), and the RPIM formulation requires the explicit construction of such a background mesh. All these reasons significantly increase the computational cost of RPIM formulations.

#### 3.1.2. Homogenised Material Properties

Next, since the aim was to investigate the influence of the PUF volume fraction ($v_{f}$) on the final homogenised mechanical properties, several decreasing volume fractions were considered. Thus, according to Equation (Equation 2) and considering the volume fractions presented in Table 4, distinct corresponding radii for the circular hole were obtained, as seen in Table 4.

The results of the convergence test, Figure 3, show that a nodal mesh with a nodal density corresponding to the nodal mesh of about 4000 nodes for $v_{f}=0.75$ was enough, providing results very close to the theoretical converged value. Thus, the following analysed RVEs were all discretised with nodal meshes containing a proportional mesh density, as can be observed in Figure 8.

The same geometry presented in Figure 1, with L=D=10 mm and bulk PUF mechanical properties, was considered: E=171.43 MPa, G=57.810 MPa, and ν=0.30. The homogenised mechanical properties for each volume fraction $v_{f}$ were once again estimated using FEM and RPIM formulations, and the obtained results are presented in Figure 9.

As Figure 9 shows, the homogenised elastic mechanical properties are dependent on the PUC volume fraction. From the graphs, it is perceptible that it is possible to overlap a polynomial curve for each homogenised elastic mechanical property as a function of the volume fraction. The results show that the FEM and RPIM formulations provided close results and followed the same trend. The most significant difference can be observed for the Poisson’s ratio, particularly for lower volume fractions. With these results, it is possible to build macroscale sandwich beam or plate models, with a PUF core, and analyse such large-scale applications using the homogenised mechanical properties of PUF with distinct volume fractions (due to the cylindrical holes).

As in the previous nodal mesh convergence study, the results obtained for the homogenised elastic mechanical properties presented in Figure 9 show that all the RPIM formulations led to very close results. They were so close that apparently all the RPIM curves overlapped each other. Such an observation can be confirmed with Table 5, in which the homogenised mechanical properties of the PUF, with distinct volume fractions, are presented for the FEM-3n, CRPIM, and MRPIM16 formulations.

Notice that for the MRPIM16 formulation, the homogenised Poisson’s ratio for $v_{f}=0.5$ was higher than ν=0.5. Such a result, also observed in other kind of foams [40] for sandwich plates, hinders the possibility to use a plane strain assumption in a macroscale application using such homogenised mechanical properties. However, 2D plane stress deformation and complete 3D deformation theories can be used. Thus, in order to use the same numerical starting point, in the macroscale analysis, the homogenised mechanical properties were only obtained with the FEM-3n.

### 3.2. Macroscale Analysis

In this section, the homogenised mechanical properties shown in Table 5 are applied to analyse a sandwich cantilever beam. In the first numerical example, homogeneous PUF cores bounded by aluminium face sheets were numerically analysed. Next, the mechanical properties of the PUF core were modified along the beam thickness, leading to an approximated functionally graded sandwich cantilever beam. In both examples, the following linear elastic isotropic mechanical properties for aluminium were considered [41]: a Young modulus of E=69.6 GPa and a Poisson’s ratio of ν=0.33. The problems here presented were analysed with five distinct formulations—FEM-4n, CRPIM and MRPIM4, MRPIM9, and MRPIM16—considering quadrilateral integration cells for the construction of the background integration mesh. Regarding the discretisation, both analysed problems used a regular mesh with 30×120 divisions, leading to a uniform regular mesh of 3751 nodes and 3600 quadrilateral elements (or, in the case of the CRPIM and MRPIM, 3600 integration cells).

#### 3.2.1. Sandwich Cantilever Beam

In the first example, the sandwich cantilever beam presented in Figure 10 was analysed. The beam, clamped in the left side, possesses the following dimensions: L=4 m, $D_{1}=D_{3}=0.1$ m, and $D_{2}=0.8$ m. In the Oz direction, an unitary thickness (H=1 m) was considered. In the right-top corner of the beam, a localized load was applied: $P=10^{6}$ N. Aiming to understand the influence on the beam’s displacement and stress fields of using PUF cores with distinct volume fractions, several analyses were performed considering for the PUF core the following six volume fractions: $v_{f}=\left\{0.5,0.6,0.7,0.8,0.9,1.0\right\}$. Then, from Table 5, the corresponding homogenised mechanical properties (obtained using the FEM-3n) were considered.

The displacement components along Ox (*u*) and along Oy (*v*) obtained in point A (represented in Figure 10) are presented in Table 6 and Table 7, respectively.

It is possible to observe that the MRPIM4 consistently presented results very close compared with the FEM-4n, with both results separated by only a 1% difference. On the other hand, the CRPIM and MRPIM16 also presented results that were extremely close, showing differences below 0.01%. Comparing the CRPIM (or the MRPIM16) with the FEM-4n, the difference between solutions increased to around 8%.

Regarding the stress field, the normal stress $\sigma_{xx}$ at point B (in the top aluminium layer) and the shear stress $\tau_{xy}$ at point C (in the PUF core at y=0.5 m) were documented, which are both represented in Figure 10. The obtained results are presented in Table 8 and Table 9.

In accordance with the results of $u_{A}$ and $v_{A}$ already discussed, it is possible to observe that once again the MRPIM4 and FEM-4n produced results very close to each other. Also, the CRPIM and MRPIM16 obtained very similar solutions. One possible reason for this results is the similarity between the nodal connectivity of the MRPIM4 and the FEM-4n, as well as the CRPIM and the MRPIM16. Both the MRPIM and FEM-4n used only four nodes to construct their shape functions at each integration point. Moreover, due to geometric coincidences, since uniform regular nodal meshes with regular quadrilateral elements (or integration cells, in the case of the CRPIM or MRPIM formulaitons) were being used, the four nodes closer to each integration point were the same, regardless of being an FEM-4n or an MRPIM formulation. For the CRPIM and MRPIM16 formulations, the background integration mesh is the same, and the shape functions of each integration point will be constructed using exactly the same 16 nodes in both formulations. Therefore, the only difference between the CRPIM and MRPIM16 is the shape parameters *c* and *p*, explaining why both formulations produced such close results.

In order to show how the normal stress $\sigma_{xx}$ varies on the clamped edge (along the beam thickness), Figure 11, Figure 12 and Figure 13 are presented. Since the magnitude of the normal stress $\sigma_{xx}$ changes significantly from the aluminium face sheets to the PUF core, in Figure 12 presents the normal stress $\sigma_{xx}$ on the PUF core, and in Figure 11 and Figure 13, the normal stress $\sigma_{xx}$ on the top and bottom aluminium layers are shown, respectively.

It is also relevant to understand how the shear stress $\tau_{xy}$ varies along the beam thickness on the clamped edge. Therefore, in Figure 14, Figure 15 and Figure 16, such variation is presented. Likewise, the normal stress $\sigma_{xx}$ and the magnitude of the shear stress $\tau_{xy}$ observed in the top and bottom aluminium layers are very different from the magnitudes in the PUF core. Therefore, once again, their representation is separated: Figure 15 presents the shear stress $\tau_{xy}$ on the PUF core, and Figure 14 and Figure 16 correspond to the shear stress $\tau_{xy}$ on the top and bottom aluminium layers, respectively.

With Figure 11 (normal stress $\sigma_{xx}$ at the top aluminium layer), it is possible to observe that the FEM-4n and MRPIM4 produced very close solutions. Another interesting observation is how close the CRPIM and MRPIM16 solutions were with the FEM-4n and MRPIM4 solutions. The main difference between these four formulations was observed only at the ends of the top aluminium layer (y∈[0.98,1.00] and y∈[0.90,0.92]). This localised difference is explained by the number of nodes inside the influence domain of both the CRPIM and MRPIM16. By assuming a fixed number of 16 nodes inside the influence domain of each integration point, corner integration points (such as the ones close to point B) will construct shape functions with larger support domain radii than other inner integration points. This effect will smooth the approximated variable (decreasing its magnitude) at these peripheral integration points [21]. The same observation can be made with Figure 13 (normal stress $\sigma_{xx}$ at the bottom aluminium layer).

Regarding the PUF core, Figure 12 shows that the CRPIM and MRPIM16 followed a similar trend. Again, the FEM-4n and MRPIM4 solutions were almost overlapped. Due to the bending effect, positive values (tensile stress) for $\sigma_{xx}$ when y>0.5 m were expected and negative values (compressive stress) when y>0.5 m. However, such an effect was not clearly observed, particularly for the CRPIM and MRPIM16 formulations, showing $\sigma_{xx}<0$ for y>0.5 m, and $\sigma_{xx}>0$ for y<0.5 m, which featured the opposite of the expected effect. Such results can be explained (once again) by the number of nodes inside the influence domain. The integration points in the PUF core, near the interface between the aluminium sheet and the PUF core, have the strong influence of the strain/stress field of the aluminium sheet. Thus, since some nodes belonging to the aluminium sheet are contained within the influence domain of those integration points (that actually belong to the PUF core), the integration points on the PUF core near the interface will show a localised perturbation on the strain/stress field. Such an effect is not visible when a lower connectivity was considered: the MRPIM4. Both the CRPIM and MRPIM16 formulations showed $\sigma_{xx}>0$ for y>0.5 m and $\sigma_{xx}<0$ for y<0.5 m, which was the expected effect. Only at the interface, due to the vicinity of the aluminium sheet, the signal was inverted.

The normal stress $\sigma_{xx}$ level in the PUF core was very low, showing the highest magnitudes observed in the top and bottom aluminium layers.

For the shear stress $\tau_{xy}$, the results show that the highest values could be found, again, at the top and bottom aluminium layers (Figure 14 and Figure 16). Concerning the PUF core, Figure 15, the distribution was approximately parabolic, as expected. The shear stress $\tau_{xy}$ maintained the trend observed for the displacement and normal stress $\sigma_{xx}$, the FEM-4n and MRPIM4s solutions, were always very close between each other, and the CRPIM and MRPIM16 results were also very close to each other. Regarding the perturbation effect observed for the normal stress $\sigma_{xx}$ near the aluminium/PUF interface, the same was also visible here; however, it was highly attenuated.

#### 3.2.2. Functionally Graded Sandwich Cantilever Beam

In the final macroscale example, two cantilever beams with aluminium face sheets and approximately functionally graded PUF cores were analysed. Thus, as shown in Figure 17, the PUF core was divided into eight layers, and the beam possessed the following dimensions: L=4 m and $D_{i}=0.1$ m for i={1,2,⋯,10}. The beam was clamped on the left side, and a localised load $P=10^{6}$ N was applied at the top-right corner.

Two distinct approximately functionally graded PUF cores were analysed (FG1 and FG2), with the mechanical properties of each layer indicated in Table 10.

Both beams were analysed with the following formulations: the FEM-4n, CRPIM, MRPIM4, MRPIM9, and MRPIM16. The Ox displacements obtained in point A (represented in Figure 17) are presented in Table 11, and the corresponding Oy displacements are shown in Table 12. As already observed in previous examples, the FEM-4n and MRPIM4, as well as the CRPIM and MRPIM16, produced similar results between each other. Thus, the difference in the magnitude of the $u_{A}$ or $v_{A}$ displacement components between the FEM-4n and MRPIM4 was about 0.04%, and between the CRPIM and MRPIM16, it was about 0.02%. However, the difference between the pair FEM-4n/MRPIM4 with the pair CRPIM/MRPIM16 was around 4%. These results show that all formulations produced approximated solutions.

Regarding the normal stress at point B and the shear stress at point C, both represented in Figure 17, the obtained results are shown in Table 13 and Table 14, respectively.

The results again show a similarity between the FEM-4n and MRPIM4 solutions, around 1% for $\sigma_{xx}$ and 2.5% for $\tau_{xy}$, and also for the solutions of the CRPIM and MRPIM16, around 0.5% for $\sigma_{xx}$ and 0.1% for $\tau_{xy}$. Such differences are in accordance with previous studies. Comparing the solutions of the FEM-4n with CRPIM, higher differences were found: about 10% for $\sigma_{xx}$ and only 5% for $\tau_{xy}$. Such a difference corroborates the previous idea that (in meshless methods with a large number of nodes inside the influence domain) integration points near a boundary (such as point B) have larger support domains, locally lowering the magnitude of the variable field (due to smoothing effects), and integration points inside the physical domain (such as point C) possess smaller support domains, allowing for sharper values.

In order to visualise the distribution of the stress field on the clamped edge, in Figure 18 and Figure 19 are shown the distribution of the normal stress $\sigma_{xx}$ and shear stress $\tau_{xy}$ along the thickness of the beam.

In Figure 18a,b,e,f, it is possible to visualise that the CRPIM and MRPIM16 values near the edge surface of the beam (y≃0.0 and y≃1.0) were smoothed. This is why the localised results of the CRPIM and MRPIM16, presented in Table 13 and Table 14, are different from the ones obtained with the FEM-4n and MRPIM4. Observing the normal stress distribution between y∈]0.9,1.0[ m and y∈]0.0,0.1[ m, it is possible to understand that all solutions were very close (with the exception of the MRPIM9). Regarding the normal stress along y∈[0.1,0.9] m, despite the similarity of pairs FEM-4n/MRPIM4 and CRPIM/MRPIM16, all solutions were acceptably close to each other.

For the shear stress $\tau_{xy}$, the results are in accordance with previous studies. Notice that at the free edges of the beam (y≃0.0 and y≃1.0), in Figure 19a,b,e,f, the solutions of the CRPIM and MRPIM16 significantly differed from the solutions of the FEM-4n and MRPIM4 (due to the size of the support domain). However, as the solution moved to the PUF core, as seen in Figure 19c,d, the results started to match each other. All solutions seemed to produce similar results.

Figure 18 and Figure 19 also allow us to understand that, regardless of the volume fraction distribution along the PUF thickness, the distinct formulations were capable of providing very similar stress distributions.

## 4. Conclusions

In this work, a full multiscale study involving sandwich beams with a polyurethane foam (PUF) core has been presented. First, the bulk PUF was modified through the inclusion of circular holes, making it possible to create PUFs with distinct volume fractions. Then, the homogenised mechanical properties of the PUF, with respect its volume fraction, were obtained using a homogenisation technique combined with finite element methods (FEMs) and four versions of the radial point interpolation method (RPIM): the classical RPIM (CRPIM) and the modified RPIM with 4, 9, and 16 nodes inside the influence domain (MRPIM4, MRPIM9, and MRPIM16, respectively). The obtained results show that RPIM formulations are capable of delivering solutions close to high-order FEM formulations, such as quadratic triangular elements (FEM-6n). However, since background triangular integration cells were considered for all RPIM formulations, to achieve variable fields with the same level of smoothness of FEM-6n, RPIM formulations require increasing the number of integration points per triangular integration cell, leading to a very high computational cost and, consequently, decreasing the overall RPIM numerical efficiency. Next, after calculating the homogenised mechanical properties of PUFs with respect their volume fractions, those mechanical properties were applied to large-scale problems: a sandwich cantilever beam with a homogeneous PUF core and a sandwich cantilever beam with an approximated functionally graded PUF core. In these macroscale examples, differently from the microscale examples, background quadrilateral integration cells were used for all RPIM formulations. Such modification made it possible to approximate significantly the RPIM and FEM results, particularly the results obtained with the MRPIM4 formulation. Regarding the vertical displacement field of the cantilever beam with a uniform core, it was observed that the solutions obtained with the CRPIM and MRPIM16 formulations had an 8% difference when compared with the FEM solution. When the comparison was made between the FEM and MRPIM4, the difference dropped to 1%. Regarding the cantilever beam with a functionally graded core, comparing the FEM solution with the CRPIM and MRPIM16 solutions (for the vertical displacement field), a 4% difference was found, and upon comparing the FEM and MRPIM4 solutions, the difference became lower: 0.04%. For the normal stress $\sigma_{xx}$, similar differences were obtained. It was found that near the domain edge, RPIM formulations using influence domains with a large number of nodes led to local lower stress values. However, stress distributions obtained with all the distinct formulations studied tended to agree along the thickness of the beam. The results obtained for the macroscale examples consistently showed that the MRPIM4 is capable of producing results very close with the FEM-4n. Such findings are very interesting, since the MRPIM4 is the most efficient RPIM formulation, as well as the one with the lower computational cost. It was also possible to observe that all RPIM formulation possessed a higher computational cost when compared with ther FEM formulations. For instance, the MRPIM4 was four times slower than the FEM, and the CRPIM and MRPIM16 were almost ten times slower than the FEM. Such results can be explained by the lack of programming optimisation of the RPIM codes, heavy preprocessing routines, and the large nodal connectivity of RPIM formulations. The results documented in this work make it possible to verify that RPIM formulation, in its classical or modified version, is a solid alternative to the FEM. The results also show that if background triangular integration cells are being used, the number of integration points per cell should be increased, leading to a not very interesting RPIM version. Therefore, for the RPIM, background quadrilateral integration cells are much more suitable. Being a preliminary study on RPIM formulations combined with multiscale analysis, its main purpose was to assess the accuracy and robustness of RPIM in this context, paving the way for its potential application in more complex scenarios involving material and geometric nonlinearities. The results here documented will make it possible to apply the obtained mechanical properties to dynamic, buckling, large deformations, and contact analyses of sandwich structures with PUF cores. The information contained in this document make it possible to design/select PUF cores with circular voids suited for specific problems. The homogenised microscale results make it possible to correlate the density of the PUF with circular voids with macroscale mechanical properties. This output can be applied to construct more efficient structural components (maximising its stiffness/weight ratio), such as airplane wings and fuselage components, wind turbine blades, automobile impact parts (to absorb impact loads), and building construction materials, amongst many other applications.

## Figures and Tables

**Figure 1 materials-17-04466-f001:**
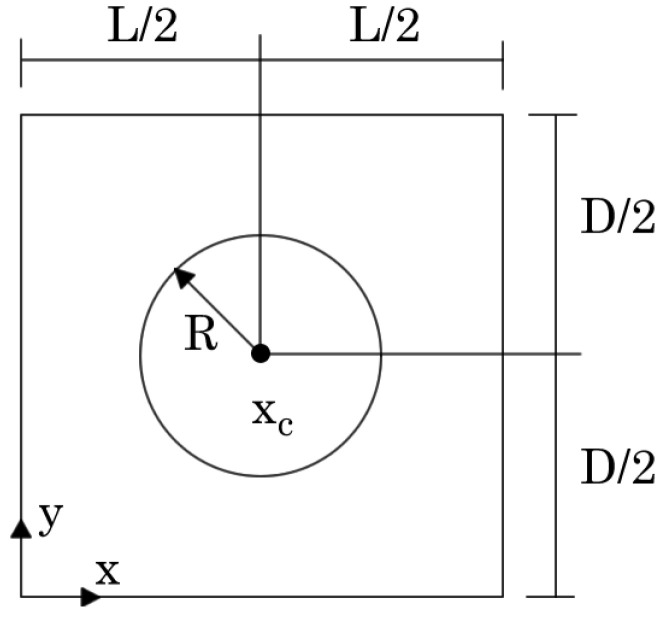
Parametric representation of the analysed RVE.

**Figure 2 materials-17-04466-f002:**
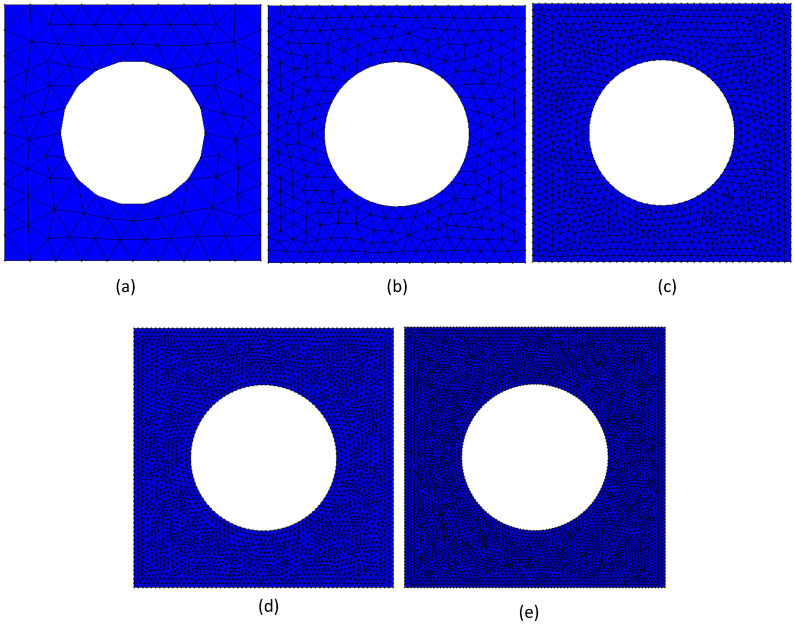
Discretisation meshes used with the FEM and RPIM analyses. (**a**) A total of 138 nodes and 234 triangular elements. (**b**) A total of483 nodes and 884 triangular elements. (**c**) A total of 1785 nodes and 3408 triangular elements. (**d**) A total of 4005 nodes and 7768 triangular elements. (**e**) A total of 6993 nodes and 13,664 triangular elements.

**Figure 3 materials-17-04466-f003:**
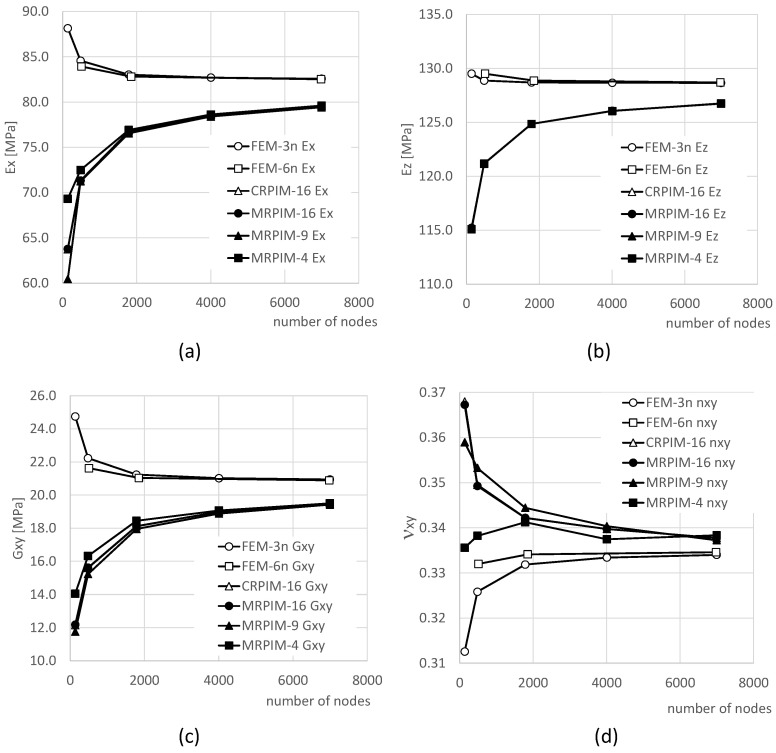
Discretisation convergence study of the following elastic mechanical properties: (**a**) $E_{x}$, (**b**) $E_{z}$, (**c**) $G_{xy}$, and (**d**) $\nu_{xy}$.

**Figure 4 materials-17-04466-f004:**
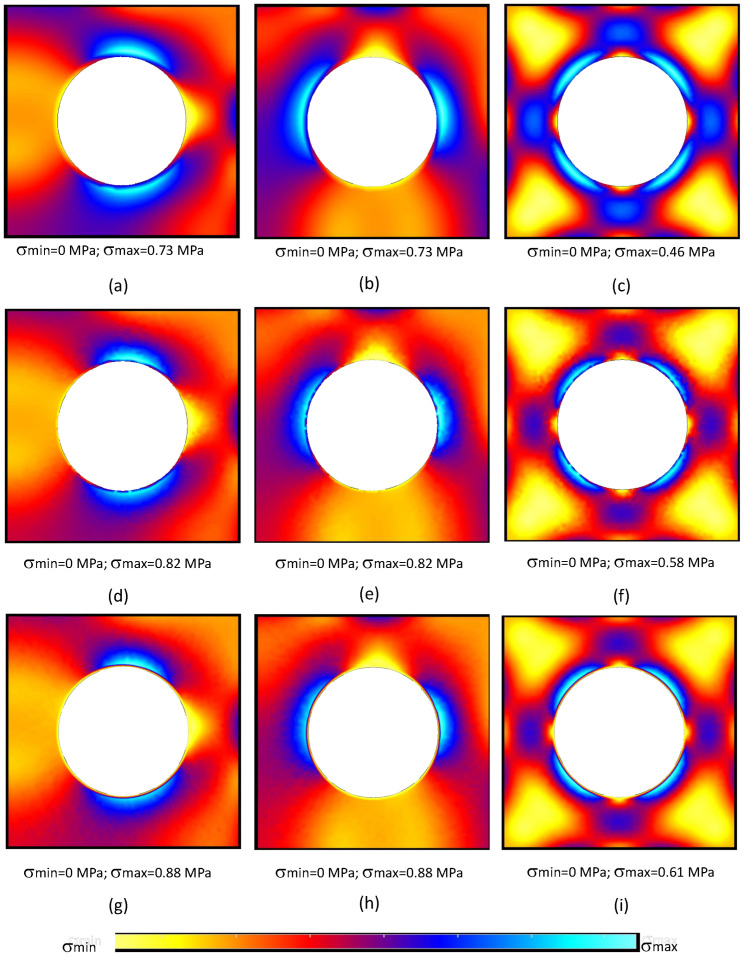
von Mises equivalent stress distribution for the following formulations: FEM-6n (**a**–**c**), FEM-3n (**d**–**f**), and CRPIM (**g**–**i**). The results corresponding to ${\tilde{f}}_{100}$ are shown in (**a**,**d**,**g**), ${\tilde{f}}_{010}$ results are shown in (**b**,**e**,**h**), and ${\tilde{f}}_{001}$ results are shown in (**c**,**f**,**i**).

**Figure 5 materials-17-04466-f005:**
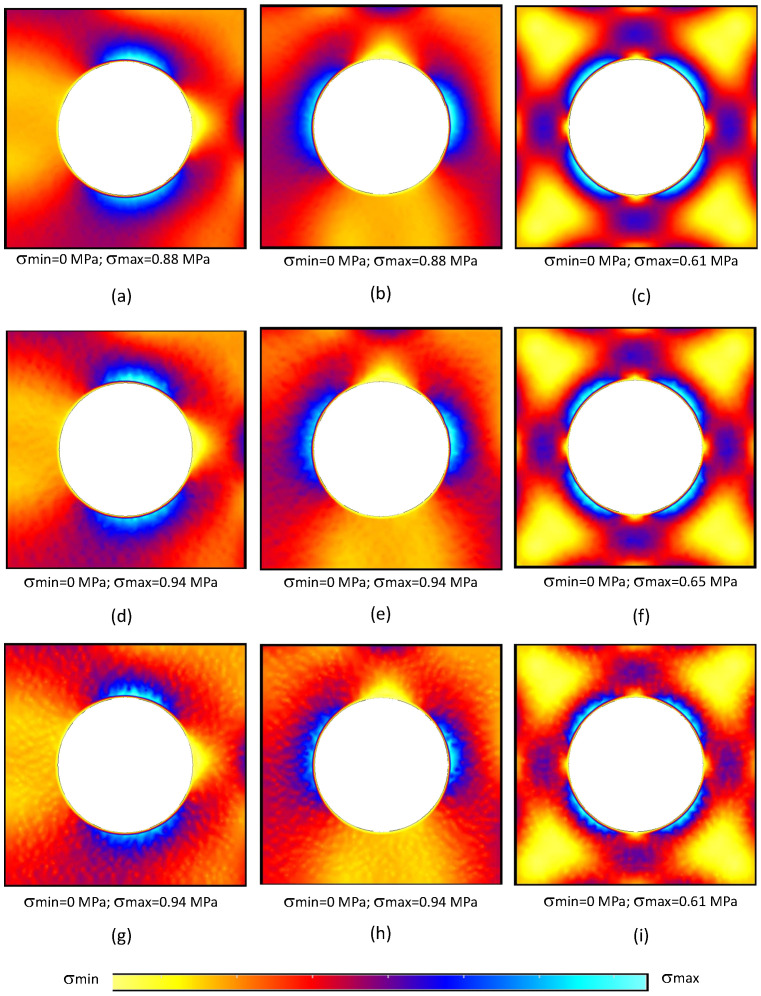
von Mises equivalent stress distribution for the following formulations: MRPIM16 (**a**–**c**), MRPIM9 (**d**–**f**), and MRPIM4 (**g**–**i**). The results corresponding to ${\tilde{f}}_{100}$ are shown in (**a**,**d**,**g**), ${\tilde{f}}_{010}$ results are shown in (**b**,**e**,**h**), and ${\tilde{f}}_{001}$ results are shown in (**c**,**f**,**i**).

**Figure 6 materials-17-04466-f006:**
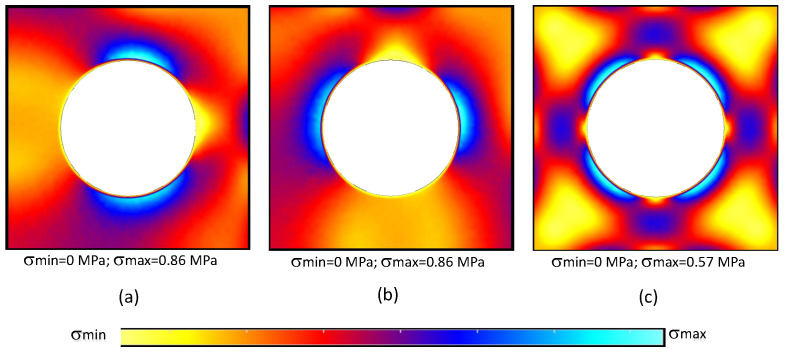
von Mises equivalent stress distribution obtained with CRPIM considering a higher-order integration scheme: (**a**) ${\tilde{f}}_{100}$, (**b**) ${\tilde{f}}_{010}$, and (**c**) ${\tilde{f}}_{001}$.

**Figure 7 materials-17-04466-f007:**
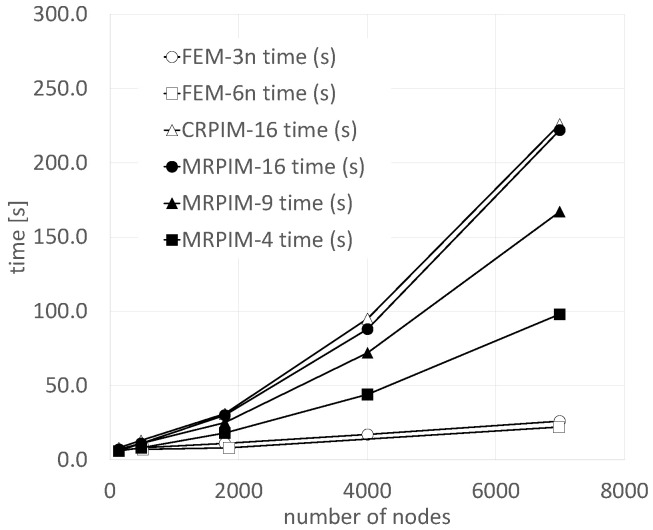
Overall computational cost of FEM and RPIM formulations.

**Figure 8 materials-17-04466-f008:**
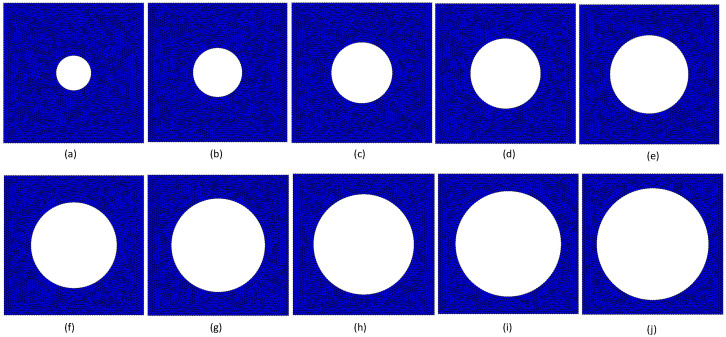
Discretisation meshes used with the FEM and RPIM analyses. (**a**) $v_{f}=0.95$; 5073 nodes and 9839 triangular elements. (**b**) $v_{f}=0.90$; 4807 nodes and 9321 triangular elements. (**c**) $v_{f}=0.85$; 4539 nodes and 8803 triangular elements. (**d**) $v_{f}=0.80$; 4273 nodes and 8285 triangular elements. (**e**) $v_{f}=0.75$; 4005 nodes and 7767 triangular elements. (**f**) $v_{f}=0.70$; 3739 nodes and 7249 triangular elements. (**g**) $v_{f}=0.65$; 3472 nodes and 6732 triangular elements. (**h**) $v_{f}=0.60$; 3205 nodes and 6214 triangular elements. (**i**) $v_{f}=0.55$; 2937 nodes and 5696 triangular elements. (**j**) $v_{f}=0.50$; 2671 nodes and 5178 triangular elements.

**Figure 9 materials-17-04466-f009:**
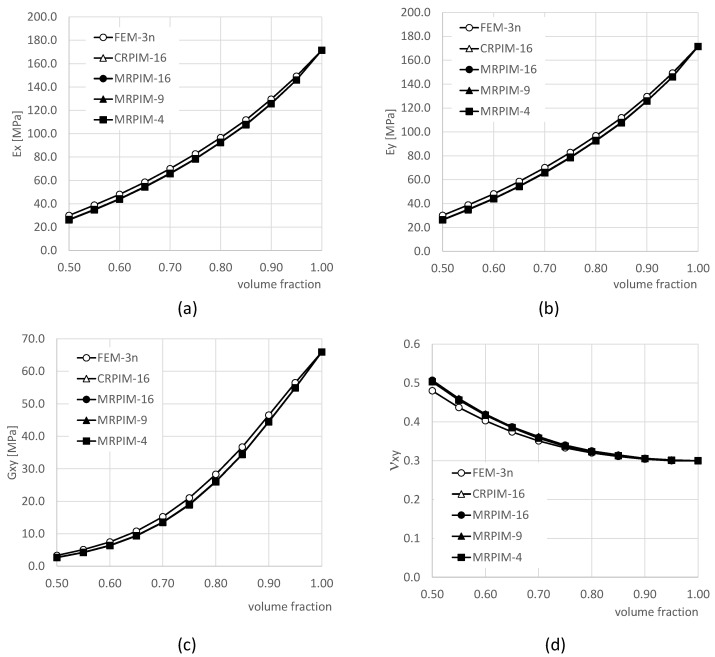
Influence of the volume fraction $v_{f}$ on the homogenised elastic mechanical properties: (**a**) $E_{x}$, (**b**) $E_{y}$, (**c**) $G_{xy}$, and (**d**) $\nu_{xy}$.

**Figure 10 materials-17-04466-f010:**
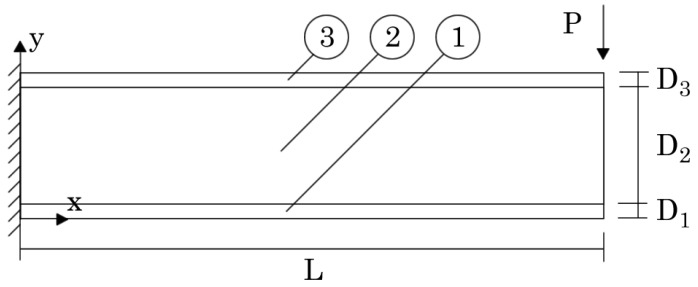
Sandwich cantilever beam with aluminium face sheets and PUF core. Three material domains were assumed: aluminium top-face sheet (3), PUF-core (2), and aluminium bottom face sheet (1).

**Figure 11 materials-17-04466-f011:**
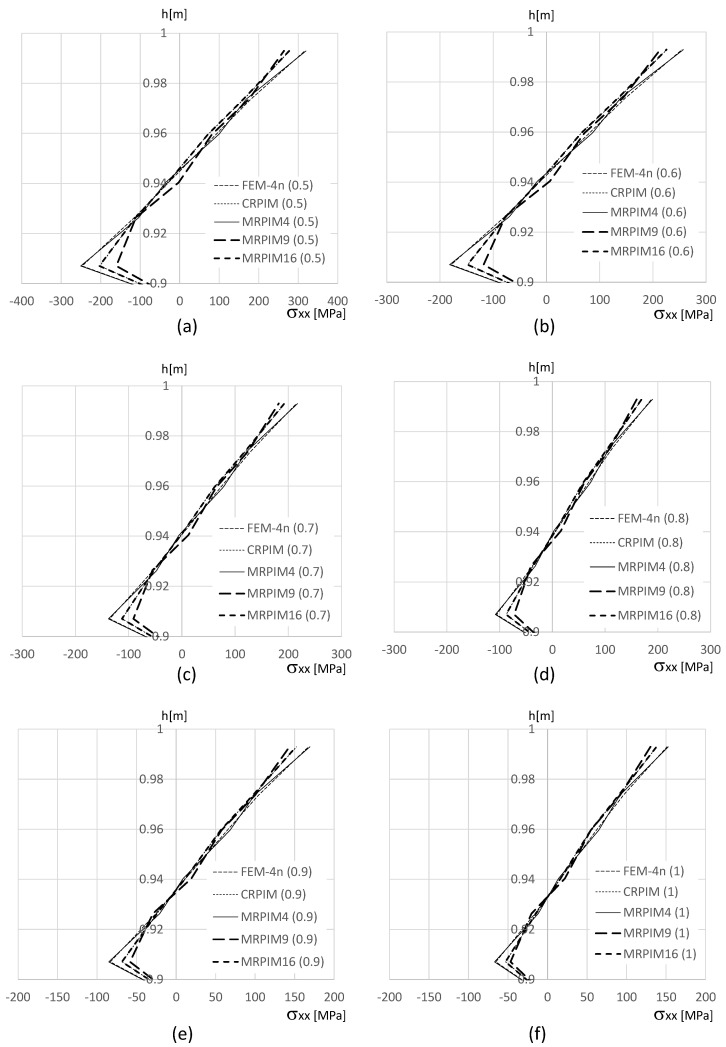
Variation in the normal stress $\sigma_{xx}$ along the aluminium top face sheet for distinct PUF cores. (**a**) $v_{f}=0.5$. (**b**) $v_{f}=0.6$. (**c**) $v_{f}=0.7$. (**d**) $v_{f}=0.8$. (**e**) $v_{f}=0.9$. (**f**) $v_{f}=1.0$.

**Figure 12 materials-17-04466-f012:**
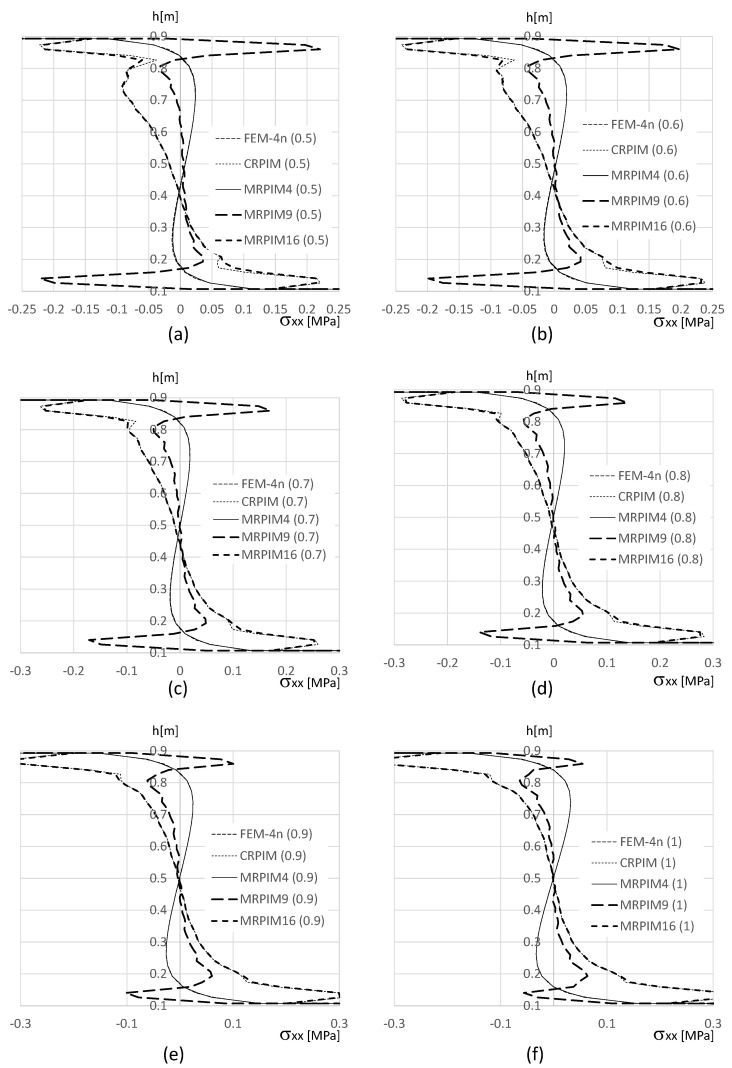
Variation in the normal stress $\sigma_{xx}$ along the PUF core for distinct PUF cores. (**a**) $v_{f}=0.5$. (**b**) $v_{f}=0.6$. (**c**) $v_{f}=0.7$. (**d**) $v_{f}=0.8$. (**e**) $v_{f}=0.9$. (**f**) $v_{f}=1.0$.

**Figure 13 materials-17-04466-f013:**
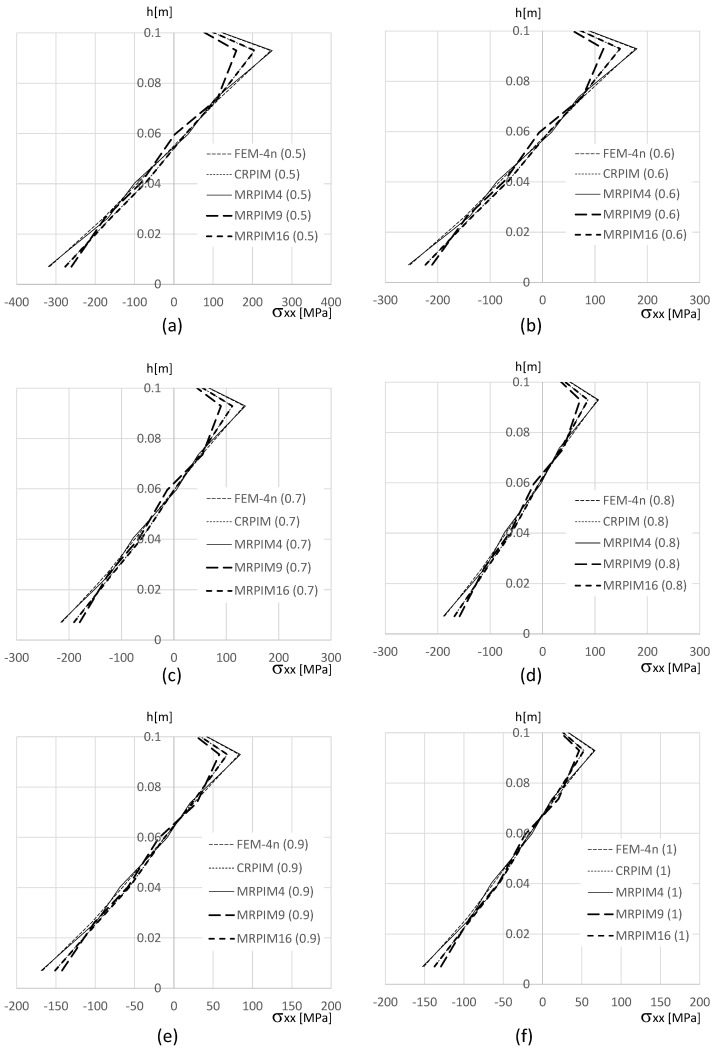
Variation in the normal stress $\sigma_{xx}$ along the aluminium bottom face sheet for distinct PUF cores. (**a**) $v_{f}=0.5$. (**b**) $v_{f}=0.6$. (**c**) $v_{f}=0.7$. (**d**) $v_{f}=0.8$. (**e**) $v_{f}=0.9$. (**f**) $v_{f}=1.0$.

**Figure 14 materials-17-04466-f014:**
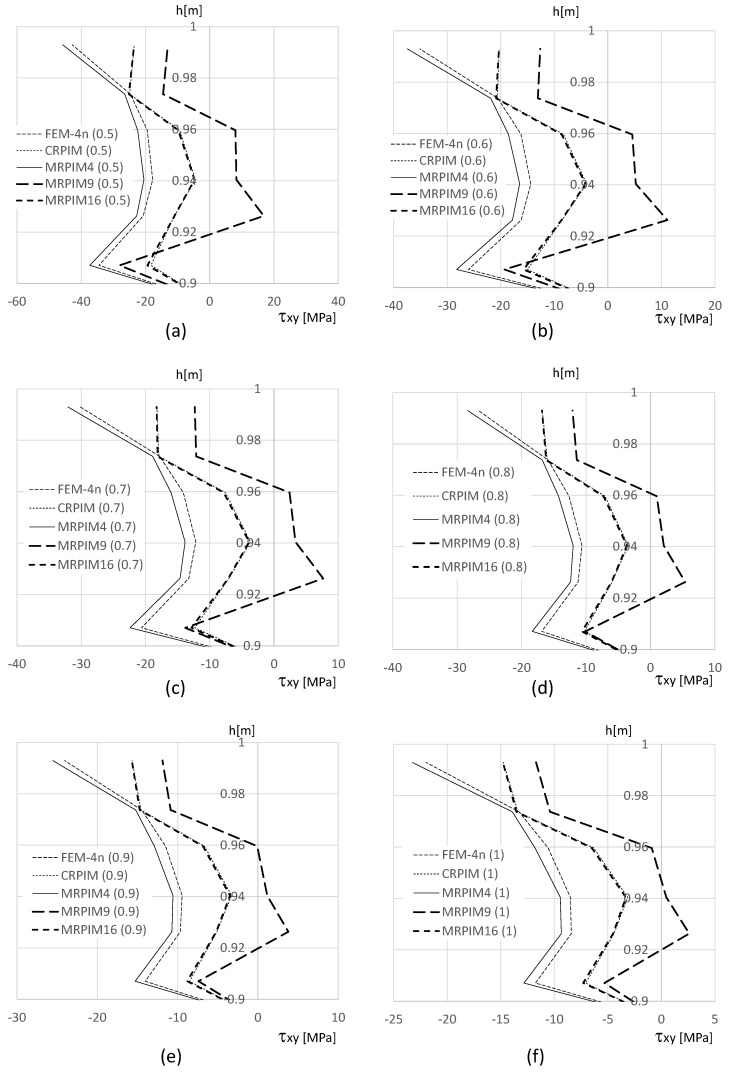
Variation in the shear stress $\tau_{xy}$ along the aluminium top face sheet for distinct PUF cores. (**a**) $v_{f}=0.5$. (**b**) $v_{f}=0.6$. (**c**) $v_{f}=0.7$. (**d**) $v_{f}=0.8$. (**e**) $v_{f}=0.9$. (**f**) $v_{f}=1.0$.

**Figure 15 materials-17-04466-f015:**
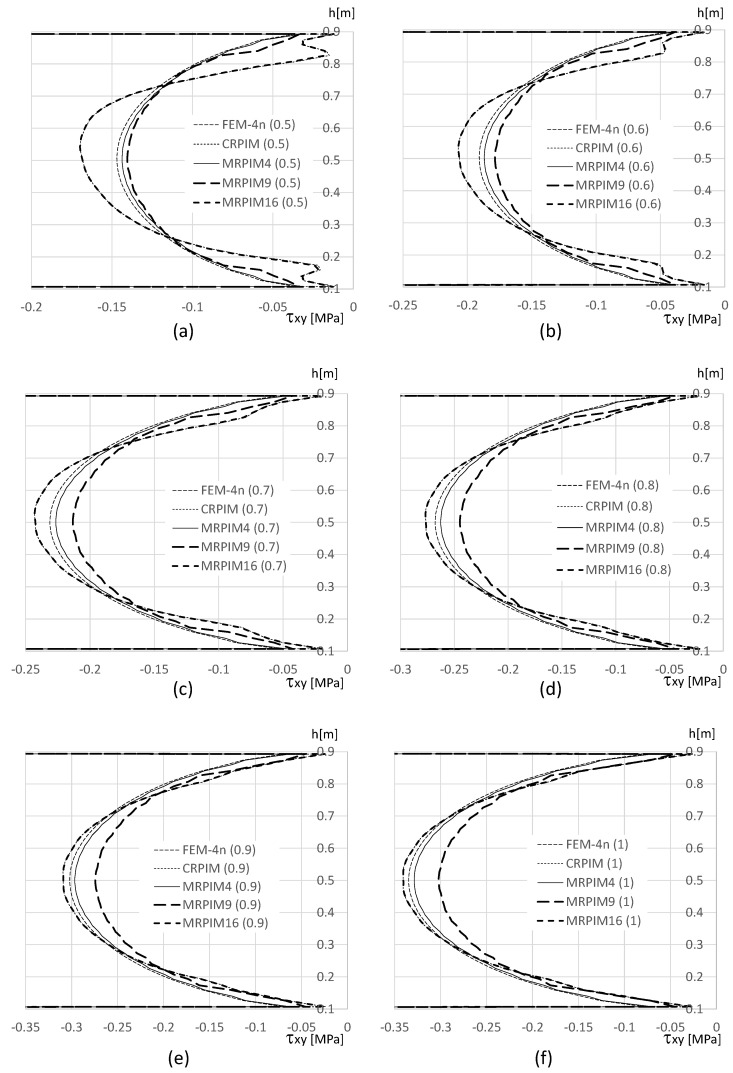
Variation in the shear stress $\tau_{xy}$ along the PUF core for distinct PUF cores. (**a**) $v_{f}=0.5$. (**b**) $v_{f}=0.6$. (**c**) $v_{f}=0.7$. (**d**) $v_{f}=0.8$. (**e**) $v_{f}=0.9$. (**f**) $v_{f}=1.0$.

**Figure 16 materials-17-04466-f016:**
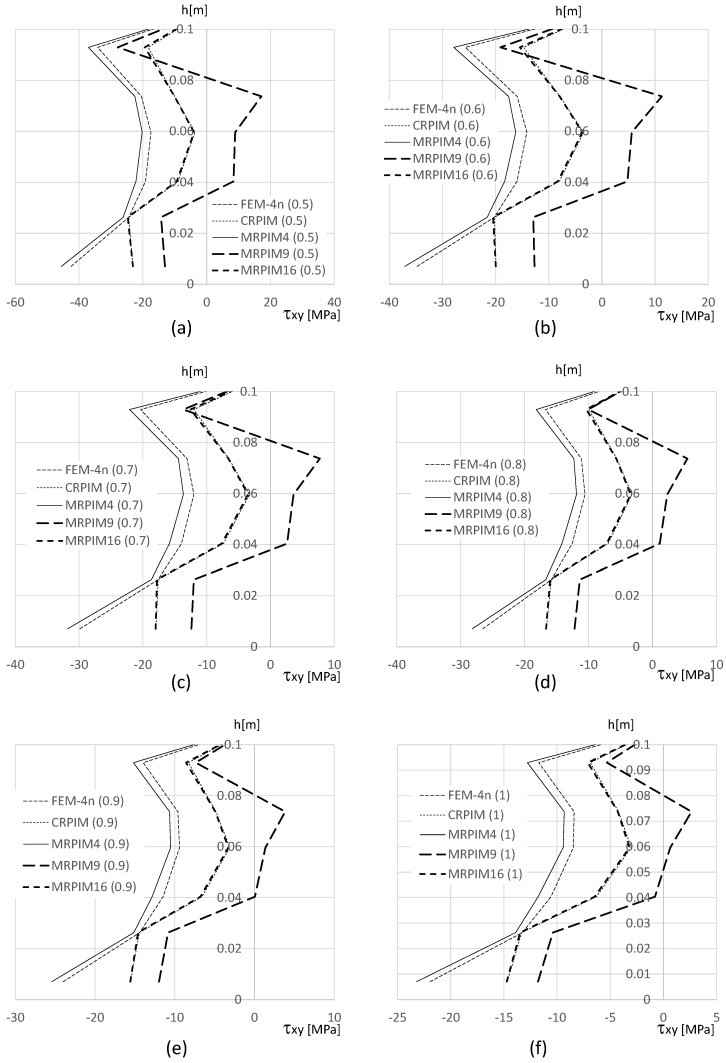
Variation in the shear stress $\tau_{xy}$ along the aluminium bottom face sheet for distinct PUF cores. (**a**) $v_{f}=0.5$. (**b**) $v_{f}=0.6$. (**c**) $v_{f}=0.7$. (**d**) $v_{f}=0.8$. (**e**) $v_{f}=0.9$. (**f**) $v_{f}=1.0$.

**Figure 17 materials-17-04466-f017:**
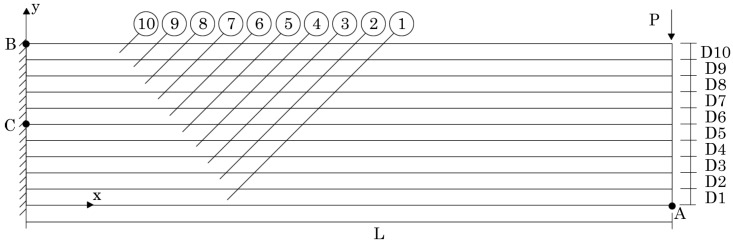
Sandwich cantilever beam with aluminium face sheets and a functionally graded PUF core. Ten material domains were assumed: aluminium top-face sheet (10), PUF-cores with potential distinct densities (2–9), and aluminium bottom face sheet (1).

**Figure 18 materials-17-04466-f018:**
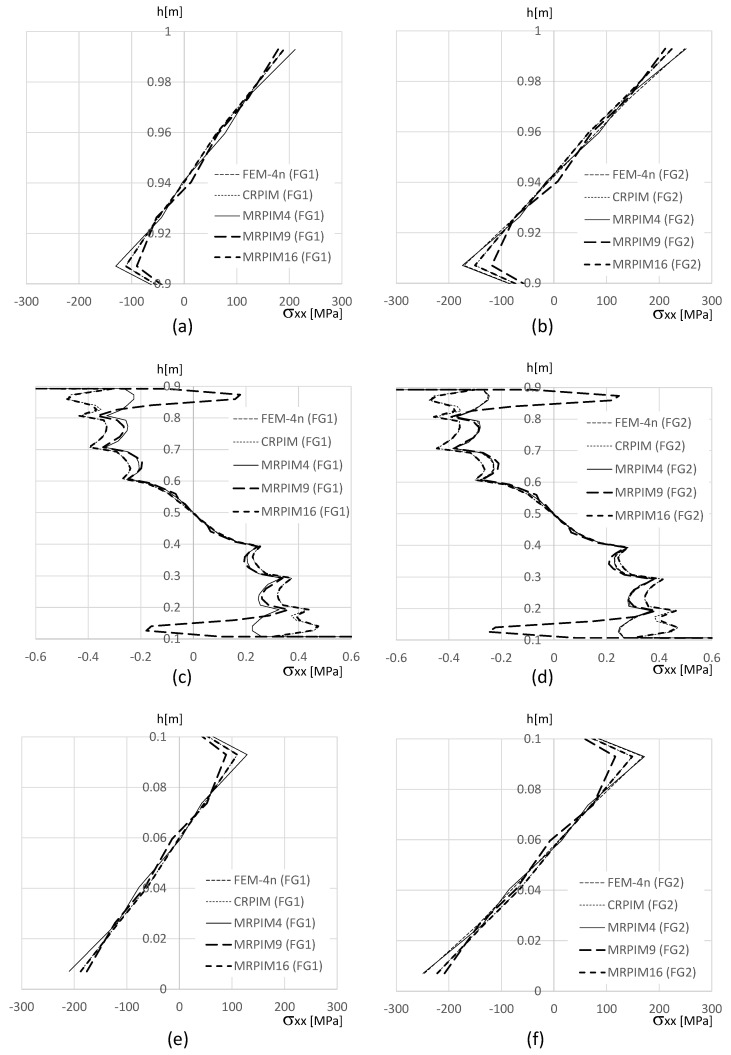
Variation in the normal stress $\sigma_{xx}$ along the beam thickness. (**a**) FG1 beam for y∈[0.9,1.0] m. (**b**) FG2 beam for y∈[0.9,1.0] m. (**c**) FG1 beam for y∈[0.1,0.9] m. (**d**) FG2 beam for y∈[0.1,0.9] m. (**e**) FG1 beam for y∈[0.0,0.1] m. (**f**) FG2 beam for y∈[0.0,0.1] m.

**Figure 19 materials-17-04466-f019:**
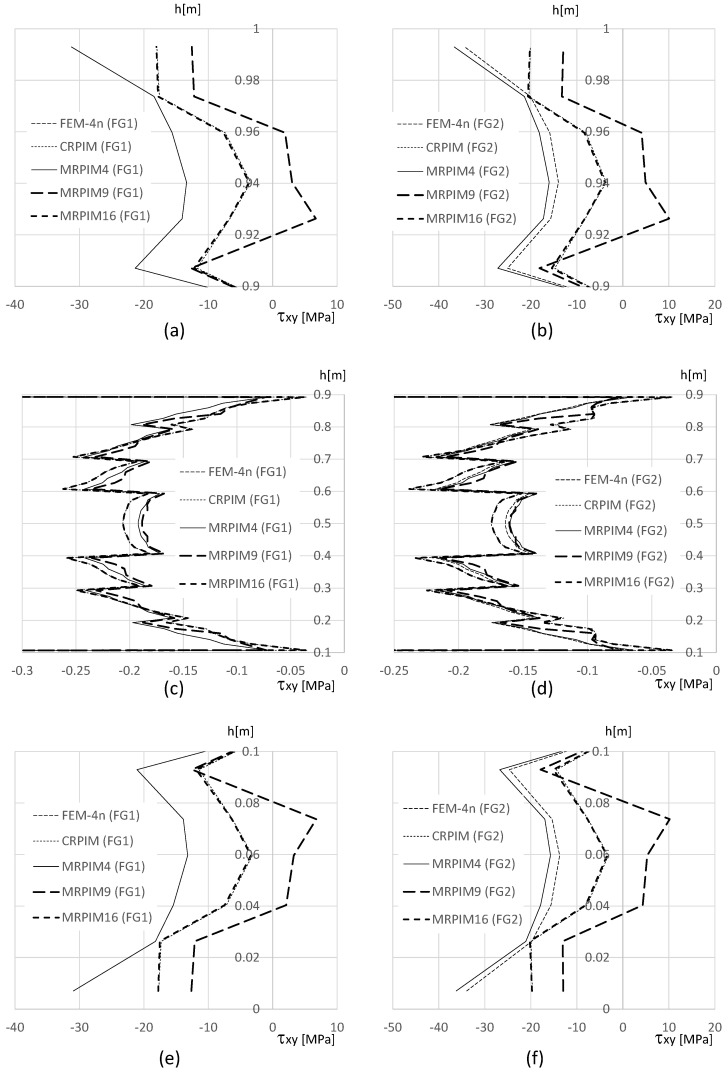
Variatio n in the shear stress $\tau_{xy}$ along the beam thickness. (**a**) FG1 beam for y∈[0.9,1.0] m. (**b**) FG2 beam for y∈[0.9,1.0] m. (**c**) FG1 beam for y∈[0.1,0.9] m. (**d**) FG2 beam for y∈[0.1,0.9] m. (**e**) FG1 beam for y∈[0.0,0.1] m. (**f**) FG2 beam for y∈[0.0,0.1] m.

**Table 1 materials-17-04466-t001:** Elastic mechanical properties of the RVE with $v_{f}=0.75$.

	Nodes	$E_{x}$ [MPa]	$E_{y}$ [MPa]	$E_{z}$ [MPa]	$G_{\mathrm{xy}}$ [MPa]	$\nu_{\mathrm{xy}}$	$\nu_{\mathrm{yx}}$
FEM-3n	138	88.155	88.251	129.508	24.745	0.313	0.313
	483	84.571	84.575	128.874	22.225	0.326	0.326
	1785	83.033	83.044	128.700	21.233	0.332	0.332
	4005	82.702	82.702	128.669	21.019	0.333	0.333
	6993	82.579	82.576	128.658	20.940	0.334	0.334
FEM-6n	509	83.924	83.922	129.508	21.621	0.332	0.332
	1849	82.808	82.808	128.874	21.033	0.334	0.334
	6977	82.519	82.519	128.700	20.887	0.335	0.335
CRPIM	138	63.739	62.498	115.229	12.147	0.368	0.368
	483	71.286	71.455	121.172	15.602	0.350	0.350
	1785	76.760	76.759	124.849	18.115	0.342	0.342
	4005	78.572	78.544	126.072	18.988	0.340	0.340
	6993	79.525	79.634	126.750	19.454	0.338	0.338
MRPIM16	138	63.782	62.556	115.227	12.175	0.367	0.367
	483	71.329	71.505	121.171	15.624	0.349	0.349
	1785	76.765	76.764	124.849	18.119	0.342	0.342
	4005	78.575	78.547	126.072	18.991	0.340	0.340
	6993	79.527	79.635	126.750	19.456	0.338	0.338
MRPIM9	138	60.443	62.784	115.220	11.748	0.359	0.359
	483	71.238	70.952	121.163	15.236	0.353	0.353
	1785	76.553	76.448	124.849	17.954	0.344	0.344
	4005	78.399	78.462	126.069	18.885	0.340	0.340
	6993	79.425	79.626	126.735	19.422	0.337	0.337
MRPIM4	138	69.318	67.072	115.098	14.046	0.336	0.336
	483	72.491	72.916	121.178	16.323	0.338	0.338
	1785	76.919	76.985	124.853	18.445	0.341	0.341
	4005	78.629	78.786	126.052	19.068	0.337	0.337
	6993	79.604	79.649	126.755	19.502	0.338	0.338

**Table 2 materials-17-04466-t002:** Elastic constitutive constants of the RVE with $v_{f}=0.75$. $C_{ij}$ in MPa.

	Nodes	$C_{11}$	$C_{12}$	$C_{21}$	$C_{22}$	$C_{33}$	$C_{44}$
FEM-3n	138	111.632	44.507	44.507	111.762	157.624	24.745
	483	107.940	44.155	44.155	107.946	156.251	22.225
	1785	106.384	44.046	44.046	106.400	155.779	21.233
	4005	106.072	44.049	44.049	106.072	155.691	21.019
	6993	105.958	44.054	44.054	105.955	155.660	20.940
FEM-6n	509	107.609	44.601	44.601	107.605	156.905	21.621
	1849	106.284	44.214	44.214	106.283	155.963	21.033
	6977	105.932	44.100	44.100	105.932	155.706	20.887
CRPIM	138	82.742	35.569	34.909	80.928	136.243	12.146
	483	91.913	39.159	39.166	92.144	144.787	15.603
	1785	98.744	41.557	41.570	98.746	150.106	18.115
	4005	100.996	42.339	42.343	100.959	151.869	18.988
	6993	102.141	42.744	42.797	102.301	152.853	19.454
MRPIM16	138	82.745	35.526	34.887	80.957	136.240	12.173
	483	91.951	39.158	39.169	92.192	144.794	15.625
	1785	98.749	41.556	41.570	98.750	150.106	18.119
	4005	100.998	42.338	42.343	100.962	151.870	18.991
	6993	102.144	42.745	42.797	102.302	152.854	19.456
MRPIM9	138	78.233	34.658	34.098	81.318	135.718	11.745
	483	92.082	39.334	39.436	91.713	144.803	15.237
	1785	98.646	41.664	41.680	98.506	150.095	17.954
	4005	100.834	42.372	42.405	100.927	151.860	18.885
	6993	101.966	42.674	42.889	102.283	152.838	19.422
MRPIM4	138	87.672	34.972	35.764	84.823	137.061	14.043
	483	92.616	38.609	38.546	93.179	144.838	16.325
	1785	98.892	41.566	41.534	98.976	150.137	18.445
	4005	100.888	42.183	42.400	101.145	151.867	19.068
	6993	102.303	42.841	42.842	102.364	152.886	19.502

**Table 3 materials-17-04466-t003:** Elastic homogenised mechanical properties and constitutive constants of the RVE with $v_{f}=0.75$ obtained with CRPIM considering a higher-order integration scheme. $C_{ij}$ in MPa.

	$E_{x}$ [MPa]	$E_{y}$ [MPa]	$E_{z}$ [MPa]	$G_{xy}$ [MPa]	$\nu_{xy}$	$\nu_{yx}$
CRPIM*	79.541	79.636	126.751	19.468	0.338	0.338
	$C_{11}$	$C_{12}$	$C_{21}$	$C_{22}$	$C_{33}$	$C_{44}$
CRPIM*	102.166	42.746	42.786	102.301	152.855	19.468

**Table 4 materials-17-04466-t004:** Relation between the volume fraction and the circular hole radius.

$v_{f}$	1.000	0.950	0.900	0.850	0.800	0.750	0.700	0.650	0.600	0.550	0.500
R [mm]	0.000	1.262	1.784	2.185	2.523	2.821	3.090	3.338	3.568	3.785	3.989

**Table 5 materials-17-04466-t005:** Homogenised mechanical properties with respect to volume fraction.

	$v_{f}$	$E_{x}$ [MPa]	$E_{y}$ [MPa]	$E_{z}$ [MPa]	$G_{\mathrm{xy}}$ [MPa]	$\nu_{\mathrm{xy}}$	$\nu_{\mathrm{yx}}$
FEM-3n	1.00	171.430	171.430	171.430	65.935	0.300	0.300
	0.95	149.211	149.212	162.913	56.481	0.300	0.300
	0.90	129.549	129.546	154.408	46.524	0.304	0.304
	0.85	111.759	111.757	145.651	36.680	0.311	0.311
	0.80	96.673	96.677	137.288	28.290	0.320	0.320
	0.75	82.702	82.702	128.669	21.019	0.333	0.333
	0.70	70.094	70.102	120.084	15.235	0.351	0.351
	0.65	58.514	58.513	111.435	10.757	0.374	0.374
	0.60	48.078	48.083	102.884	7.458	0.403	0.403
	0.55	38.870	38.865	94.580	5.110	0.437	0.437
	0.50	30.106	30.106	85.801	3.332	0.480	0.480
CRPIM	1.00	171.430	171.430	171.430	65.935	0.300	0.300
	0.95	146.086	146.025	161.717	54.923	0.300	0.300
	0.90	125.863	125.848	152.774	44.458	0.304	0.304
	0.85	107.790	107.747	143.651	34.398	0.311	0.311
	0.80	92.590	92.555	134.992	26.060	0.320	0.320
	0.75	78.572	78.544	126.072	18.988	0.333	0.333
	0.70	66.000	65.923	117.252	13.508	0.351	0.351
	0.65	54.529	54.436	108.420	9.355	0.374	0.374
	0.60	44.137	44.071	99.648	6.352	0.403	0.403
	0.55	34.952	34.967	91.113	4.247	0.437	0.437
	0.50	26.342	26.335	82.151	2.691	0.480	0.480
MRPIM16	1.00	171.430	171.430	171.430	65.935	0.300	0.300
	0.95	146.087	146.027	161.717	54.926	0.301	0.301
	0.90	125.864	125.850	152.774	44.460	0.306	0.306
	0.85	107.793	107.750	143.651	34.402	0.314	0.314
	0.80	92.592	92.559	134.992	26.063	0.325	0.325
	0.75	78.575	78.547	126.072	18.991	0.340	0.340
	0.70	66.003	65.927	117.252	13.510	0.360	0.360
	0.65	54.532	54.440	108.420	9.357	0.387	0.387
	0.60	44.141	44.075	99.647	6.354	0.419	0.419
	0.55	34.957	34.972	91.113	4.248	0.458	0.458
	0.50	26.346	26.339	82.151	2.692	0.507	0.507

**Table 6 materials-17-04466-t006:** Displacement obtained at point A along direction Ox for sandwich cantilever beams with PUF cores with distinct volume fractions. *u* in [mm].

Vol. Frac.	FEM-4n	CRPIM	MRPIM4	MRPIM9	MRPIM16
0.5	−5.194	−4.785	−5.169	−4.895	−4.782
0.6	−3.642	−3.382	−3.637	−3.441	−3.381
0.7	−2.831	−2.657	−2.831	−2.694	−2.657
0.8	−2.369	−2.248	−2.369	−2.273	−2.245
0.9	−2.078	−1.991	−2.079	−2.009	−1.991
1.0	−1.883	−1.819	−1.884	−1.833	−1.819

**Table 7 materials-17-04466-t007:** Displacement obtained at point A along direction Oy for sandwich cantilever beams with PUF cores with distinct volume fractions. *v* in [mm].

Vol. Frac.	FEM-4n	CRPIM	MRPIM4	MRPIM9	MRPIM16
0.5	−284.043	−261.657	−281.414	−270.417	−261.661
0.6	−185.148	−170.287	−183.955	−175.344	−170.282
0.7	−130.169	−119.857	−129.539	−123.084	−119.847
0.8	−96.882	−89.406	−96.509	−91.639	−89.398
0.9	−74.711	−69.153	−74.478	−70.780	−69.144
1.0	−58.909	−54.722	−58.756	−55.952	−54.715

**Table 8 materials-17-04466-t008:** Normal stress $\sigma_{xx}$ obtained at point B for sandwich cantilever beams with PUF cores with distinct volume fractions. $\sigma_{xx}$ in MPa.

Vol. Frac.	FEM-4n	CRPIM	MRPIM4	MRPIM9	MRPIM16
0.5	−316.024	−277.339	−319.636	−260.559	−276.372
0.6	−252.946	−224.091	−255.732	−210.699	−223.372
0.7	−213.791	−191.043	−216.064	−179.627	−190.476
0.8	−187.105	−168.441	−189.052	−158.326	−167.976
0.9	−167.088	−151.419	−168.806	−142.262	−151.030
1.0	−151.028	−137.711	−152.571	−129.310	−137.382

**Table 9 materials-17-04466-t009:** Shear stress $\tau_{xy}$ obtained at point C for sandwich cantilever beams with PUF cores with distinct volume fractions. $\tau_{xy}$ in kPa.

Vol. Frac.	FEM-4n	CRPIM	MRPIM4	MRPIM9	MRPIM16
0.5	−146.937	−168.516	−143.741	−140.503	−168.624
0.6	−190.784	−206.409	−186.841	−178.593	−206.404
0.7	−231.156	−242.687	−226.565	−213.384	−242.622
0.8	−267.833	−276.681	−262.707	−244.885	−276.580
0.9	−302.068	−309.140	−296.495	−274.212	−309.015
1.0	−334.716	−340.668	−328.765	−302.107	−340.528

**Table 10 materials-17-04466-t010:** Mechanical properties of each layer.

Beam	Layer	Vol. Frac.	Material	E [MPa]	ν
FG1	D1	1	aluminium	69600	0.330
	D2	0.9	PUF	129.549	0.304
	D3	0.8	PUF	96.673	0.320
	D4	0.7	PUF	70.094	0.351
	D5	0.6	PUF	48.078	0.403
	D6	0.6	PUF	48.078	0.403
	D7	0.7	PUF	70.094	0.351
	D8	0.8	PUF	96.673	0.320
	D9	0.9	PUF	129.549	0.304
	D10	1	aluminium	69600	0.330
FG2	D1	1	aluminium	69600	0.330
	D2	0.8	PUF	96.673	0.320
	D3	0.7	PUF	70.094	0.351
	D4	0.6	PUF	48.078	0.403
	D5	0.5	PUF	30.106	0.480
	D6	0.5	PUF	30.106	0.480
	D7	0.6	PUF	48.078	0.403
	D8	0.7	PUF	70.094	0.351
	D9	0.8	PUF	96.673	0.320
	D10	1	aluminium	69600	0.330

**Table 11 materials-17-04466-t011:** Displacement obtained at point A along direction Ox for sandwich cantilever beams with functionally graded PUF cores. *u* in [mm].

Vol. Frac.	FEM-4n	CRPIM	MRPIM4	MRPIM9	MRPIM16
FG1	−2.703	−2.607	−2.704	−2.624	−2.606
FG2	−3.495	−3.361	−3.492	−3.386	−3.359

**Table 12 materials-17-04466-t012:** Displacement obtained at point A along direction Oy for sandwich cantilever beams with functionally graded PUF cores. *v* in [mm].

Vol. Frac.	FEM-4n	CRPIM	MRPIM4	MRPIM9	MRPIM16
FG1	−122.141	−116.725	−121.579	−118.573	−116.706
FG2	−176.362	−169.300	−175.271	−172.018	−169.269

**Table 13 materials-17-04466-t013:** Normal stress $\sigma_{xx}$ obtained at point B for sandwich cantilever beams with functionally graded PUF cores. $\sigma_{xx}$ in MPa.

Vol. Frac.	FEM-4n	CRPIM	MRPIM4	MRPIM9	MRPIM16
FG1	−207.505	−188.319	−209.727	−176.261	−187.781
FG2	−246.623	−222.737	−249.362	−208.174	−222.048

**Table 14 materials-17-04466-t014:** Shear stress $\tau_{xy}$ obtained at point B for sandwich cantilever beams with functionally graded PUF cores. $\tau_{xy}$ in kPa.

Vol. Frac.	FEM-4n	CRPIM	MRPIM4	MRPIM9	MRPIM16
FG1	−196.320	−206.232	−191.908	−188.425	−206.021
FG2	−163.736	−174.509	−159.874	−160.641	−174.319

## Data Availability

The original contributions presented in the study are included in the article, further inquiries can be directed to the corresponding author.

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
