# Peer review of "Multiscale Analysis of Sandwich Beams with Polyurethane Foam Core: A Comparative Study of Finite Element Methods and Radial Point Interpolation Method"

_materials, 2024, doi:10.3390/ma17184466_

Round 1

Reviewer 1 Report

Comments and Suggestions for Authors

Please check attachment.

Comments on the Quality of English Language English vocabulary needs to be more concise

Author Response

Dear reviewer 1,

Thank very much for your constructive comments and suggestions, which allowed to improve the manuscript. Next, I will address each comment and within the manuscript, the reviewer can find the modified text in blue and red.

Comment 1:

Novelty needs to be highlighted in a better way. At present, the introduction section is too lengthy and complex, and cannot effectively explain the novelty of the current work.

Answer: This work deals presents a full computational framework to mechanically analyse PUF materials (without assuming non-linear material or large strain regimes). Therefore, it is natural that the first section becomes larger than average. It was structured in the following manner: (1) general introduction to the topic, (2) most relevant exact theories for analysing sandwich structures, (3) homogenization techniques for multiphase materials (such as PUF materials), (4) discretization techniques for mechanical analysis of sandwich structures, (5) meshless methods (the focus of this work) and its combination with composite and sandwich structures, (6) novelty, aims and scope of the present manuscript, and (7) manuscript layout.

Regarding the novelty of the manuscript, in the introduction section it was highlighted a concluding text detailing the novelty of this research and its chief goals (blue text).

Comment 2:

What is the specific source of polyurethane foam (PUF) core material used? If it is self-prepared, what is the specific preparation method?

Answer: Since this paper is a computational paper, focused on the numeric simulation of sandwich structures using meshless methods, the PUF is just a theoretical material (such as the face-sheet aluminium layers). The aluminium and PUF material are virtual (not real), and therefore the PUF was not prepared. For linear elasto-static numeric simulations it is only relevant to know the elasticity modulus and Poisson’s ratio of the material (in its final configuration). The computational analysis procedure does not takes directly into account how the material is obtained/fabricated. If the process of manufacturing/transforming the material modifies its mechanical properties, such mechanical properties are then assumed in the computational analysis procedure.

In this work, the PUF mechanical properties were obtained from the literature (properly cited in the submitted manuscript):

Dhaliwal, G.; Newaz, G. Flexural Response of Degraded Polyurethane Foam Core Sandwich Beam with Initial Crack between Facesheet and Core. Materials 2020, 13, 5399

Comment 3:

Is there a significant difference between the final data analyzed and the actual data? Is there any actual test that can be compared with it?

Answer: This submitted manuscript is actually a pure numeric research. Here, two numeric techniques are compared with each other: the FEM and the RPIM (a meshless method). The FEM is the most popular numerical technique in computational mechanics, widely used to simulate all sort of structural problems. The FEM was already validated (comparing its numerical results with analytical solutions and experimental results) in numerous research works, proving to be accurate and robust. For instances:

Florence, A.; Jaswin, M.; Prakash, M.; Jayaram, S. Effect of energy-absorbing materials on the mechanical behaviour of hybrid FRP 916 honeycomb core sandwich composites. Materials Research Innovations 2019, 24, 244–255

Meshless methods (such as RPIM) are still behind FEM. Nevertheless, meshless methods are also efficient methods capable to deliver accurate results. Therefore, it is important validate meshless methods. Since experimental tests are outside the scope of the present work, comparing the RPIM with FEM allows to validate it.

Comment 4:

The data in the "3. Numerical Results" section is too much and complex, making it look confusing. We hope it can be more systematic.

Answer: Regarding the data in section 3. The results were divided in 2 sub-sections. The first subsection (3.1) deals with obtaining the homogenized mechanical properties of PUF materials with distinct circular holes (allowing for distinct densities). It is a microscale structural analysis. This first subsection is relevant because it allows to define the elasticity modulus and the Poisson’s ratio required in macro-scale analysis. Therefore, first, the best mesh must be investigated (convergence study), or else one could present inaccurate results. Then, using the best mesh density, the micro-scale homogenized mechanical properties are obtained for each PUF density.

Then, in a second subsection (3.2), the homogenized mechanical properties of the PUF are considered to analyse macroscale sandwich beams. The results focused in the way the beam bends (maximum displacements, in order to understand the structure rigidity) and the way the stress field is distributed along the thickness (in order to predict material failure). First, beams with PUF cores with constant density are analysed, and then beams with functionally graded PUF cores.

Comment 5:

English vocabulary needs to be more concise.

Answer: In my best knowledge, the text was revised and the English was corrected.

Best regards,

Jorge Belinha

Reviewer 2 Report

Comments and Suggestions for Authors

Manuscript Materials-3048663 entitled "Multiscale Analysis of Sandwich Beams with Polyurethane Foam Core: A Comparative Study of Finite Element Methods and Radial Point Interpolation Method."

In this manuscript, the authors compare the performance and the efficiency of the radial point interpolation method (RPIM) with the finite element methods (FEM), such as computational cost, accuracy and mechanical properties, and explore the meshless multiscale analysis of sandwich structures with a homogeneous or a functionally graded PU foam core. The manuscript shows interesting results for the area. However, revisions would be needed before it may be considered for publication in Materials.

My comments: 

 1. For the introduction, it is unclear why the author chose the material "Polyurethane Foam (PUF)" for the analysis. It is recommended that the author mention in the introduction the importance of this material, its uses and why it is ideal for this study.

Initially, the author only mentions "composite materials" but does not specifically mention PUF.

2. Page 16, Fig. 4. The author presents the discretization convergence study, but some points are not observed in the graphs; for example, in Fig. 4b, only 3 lines are observed in the graph for 6 parameters. It is recommended that the data presented be reviewed.

3. Page 22, Fig. 9. The author presents the Influence of the volume fraction, but some points are not observed in the graphs; for example, only 2 lines are observed in the graph for 5 parameters.

4. It is recommended that the data presented in all Figures be reviewed.

5. Mention in the conclusions where this material could be applied.

Comments on the Quality of English Language

Minor editing of English language required

Author Response

Dear reviewer 2,

Thank very much for your constructive comments and suggestions, which allowed to improve the manuscript. Next, I will address each comment and within the manuscript, the reviewer can find the modified text in blue and red.

Comment 1. For the introduction, it is unclear why the author chose the material "Polyurethane Foam (PUF)" for the analysis. It is recommended that the author mention in the introduction the importance of this material, its uses and why it is ideal for this study. Initially, the author only mentions "composite materials" but does not specifically mention PUF.

Answer: Following the reviewer suggestion, a new paragraph on PUF materials was included in the beginning of the introduction section (red text).

Regarding the mention to “composite materials”, it was initially framed into the homogenization multiscale techniques. Actually, the mentioned homogenization technique (Generalized Method of Cells (GMC)) can be applied to any two-phase material (it does not has to be composite materials). Nevertheless, in order to clarify the text, the state-of-the-art corresponding to GMC was removed from the manuscript. In the revised version only a general remark is now mentioned and cited in the introduction, allowing the reader to trace back the GMC technique in the literature (red text).

Comment 2. Page 16, Fig. 4. The author presents the discretization convergence study, but some points are not observed in the graphs; for example, in Fig. 4b, only 3 lines are observed in the graph for 6 parameters. It is recommended that the data presented be reviewed.

Answer: All the data is contained in Fig.4. The problem is that the data is almost coincident, and the line markers (and also the lines) overlap each other. I already tried to change the size of the marker (and even remove the markers), but the lines still overlap each other. For the sake of clarity, this fact is now mentioned in the manuscript text (red text), and table 1 and table 2 were extended to show how close the results really are (red text).

Comment 3. Page 22, Fig. 9. The author presents the Influence of the volume fraction, but some points are not observed in the graphs; for example, only 2 lines are observed in the graph for 5 parameters.

Answer: As in the previous comment, all the data is contained in Fig.9. The problem, once again, is that the data is almost coincident, and the lines and line markers completely overlap each other. I already tried to change the size of the marker (and even remove the markers), but the lines still overlap each other. Thus, once again, for the sake of clarity, this fact is now mentioned in the manuscript text (red text), which can be confirmed with table 5.

Comment 4. It is recommended that the data presented in all Figures be reviewed.

Answer: The data was revised, and all overlapped information is now clarified in the text (red text) and its values are explicitly shown (in corresponding tables).

Comment 5. Mention in the conclusions where this material could be applied.

Answer: Following the reviewer suggestion, in the end of conclusion section it was include a paragraph on possible applications for the proposed material (red text).

Comments on the Quality of English Language

Minor editing of English language required

Answer: In my best knowledge, the text was revised and the English was corrected.

Best regards,

Jorge Belinha

Reviewer 3 Report

Comments and Suggestions for Authors

The manuscript entitled “Multiscale Analysis of Sandwich Beams with Polyurethane Foam Core: A Comparative Study of Finite Element Methods and Radial Point Interpolation Method” focus on the use of a novel approach to simulate the behavior at a microscale and at a macroscale of sandwich beams with polyurethane foam core and aluminum skins based on a meshless method, the radial point interpolation method (RPIM). Moreover, the findings are compared with the ones obtained through traditional FEM approaches, based on mesh discretization and nodal points. The work and the obtained results are well presented. In this reviewer opinion, the paper can be accepted after some clarifications.

Introduction The analysis of the state of the art is punctual and it deeply investigates the recent trends in numerical investigation of innovative material structures. At the end of the introduction, the author could better clarify the reason of using a non conventional numerical approach to investigate the behavior at a micro and macroscale of a sandwich structure. The author could also clarify possible applications of the proposed sandwich configuration.

Material and methods

The author focus on the physical modellization behind the proposed numerical approach. The theory is well presented; too much details have been presented which do not help to follow the approach of the method. All the theoretical considerations could be placed in an Appendix section.

Results

Lots results have been presented for microscale but less for macroscale analysis. In this reviewer opinion the author should give any clarification about the absence of a physical sample to validate the proposed results. In alternative, the author shoul give any information about the elastic properties which have been considered as a reference. Moreover, the author has analyzed, at a macroscale, the only configuration of a cantilever beam.

The results could be enhanced by also simultating the condition of a simply supported beam, which is also very simple to validate with a standard three point bending test. The author should give more details about the use of two different discretization methods used respectively for the investigation at a microscale and a macroscale.

Finally, the author could give more information about the computational time of the performed simulation (as the whole duration for a single numerical convergence).

Conclusion

In the conslusion section, the author could report the most significative results in quantitative terms (numbers and differences between the two numerical approaches). Moreover, further details on the future development or applications on the proposed RPIM method should be given.

Language typos

Please correct some language typos (i.e. line 502 there is the repetition of “the”; line 596 change “bellow” in below”).

Comments on the Quality of English Language

No comments are made for the quality of English.

Author Response

Dear reviewer 3,

Thank very much for your constructive comments and suggestions, which allowed to improve the manuscript. Next, I will address each comment and within the manuscript, the reviewer can find the modified text in blue and red.

Comment 1:

Introduction: The analysis of the state of the art is punctual and it deeply investigates the recent trends in numerical investigation of innovative material structures. At the end of the introduction, the author could better clarify the reason of using a non conventional numerical approach to investigate the behavior at a micro and macroscale of a sandwich structure. The author could also clarify possible applications of the proposed sandwich configuration.

Answer: Following the reviewer suggestion, it was included in the end of “introduction” section a text clarifying the novelty and aim of this manuscript, as well as a text justifying the use of meshless methods in this particular application (blue text). Regarding the possible applications of the proposed sandwich solution, in the beginning of “introduction” section it was included a text justifying the use of PUF core, and its general application fields (red text). Additionally, in the “conclusions” section a text containing further applications is presented (red text).

Comment 2:

Material and methods: The author focus on the physical modellization behind the proposed numerical approach. The theory is well presented; too much details have been presented which do not help to follow the approach of the method. All the theoretical considerations could be placed in an Appendix section.

Answer: Following the reviewer suggestion, the all the theoretical description was placed in an Appendix section.

Comment 3:

Results: Lots results have been presented for microscale but less for macroscale analysis. In this reviewer opinion the author should give any clarification about the absence of a physical sample to validate the proposed results. In alternative, the author should give any information about the elastic properties which have been considered as a reference. Moreover, the author has analyzed, at a macroscale, the only configuration of a cantilever beam.

Answer: Regarding the first comment, this submitted manuscript is actually a pure numeric research. Here, two numeric techniques are compared with each other: the FEM and the RPIM (a meshless method). The FEM is the most popular numerical technique in computational mechanics, widely used to simulate all sort of structural problems. The FEM was already validated (comparing its numerical results with analytical solutions and experimental results) in numerous research works, proving to be accurate and robust. For instances:

Florence, A.; Jaswin, M.; Prakash, M.; Jayaram, S. Effect of energy-absorbing materials on the mechanical behaviour of hybrid FRP 916 honeycomb core sandwich composites. Materials Research Innovations 2019, 24, 244–255

Meshless methods (such as RPIM) are still behind FEM. Nevertheless, meshless methods are also efficient methods capable to deliver accurate results. Therefore, it is important validate meshless methods. Since experimental tests are outside the scope of the present work, comparing the RPIM with FEM allows to validate it. This is the reason why there are no physical samples for comparison. The elastic properties of PUF were obtained from:

Dhaliwal, G.; Newaz, G. Flexural Response of Degraded Polyurethane Foam Core Sandwich Beam with Initial Crack between Facesheet and Core. Materials 2020, 13, 5399

Comte, C.; von Stebut, J. Microprobe-type measurement of Young’s modulus and Poisson coefficient by means of depth sensing indentation and acoustic microscopy. Surface and Coatings Technology 2020, 154, 42

Both research works are cited in the submitted manuscript .

Comment 4:

The results could be enhanced by also simultating the condition of a simply supported beam, which is also very simple to validate with a standard three point bending test. The author should give more details about the use of two different discretization methods used respectively for the investigation at a microscale and a macroscale.

Answer: I understand the reviewer point of view. However, presenting the cantilever beam or the three-point bending beam produces the same results. Here is the explanation. Consider a homogeneous cantilever beam with elasticity modulus E, possessing the following dimensions Le=L, De=D and He=H, and submitted to a tip concentrated load Pe=P (as the one presented in the submitted manuscript). Its theoretical vertical displacement is equal to v=Pe*Le^3/(3*E*Iz), being Iz=He*De^3/12, which is equal to: v=P*L^3/(3*E*Iz),

Consider now a three point bending beam with elasticity modulus E (i.e., a simply supported beam with a central concentrated load). The dimensions are: Le=2*L, De=D and He=H, and the concentrated load applied at Le/2 is equal to Pe=2*P. Its theoretical vertical displacement is equal to v=Pe*Le^3/(48*E*Iz), being Iz=He*De^3/12, which is equal to: v=(2*P)*(2*L)^3/(48*E*Iz) = P*L^3/(3*E*Iz). Notice that it is the same value of the cantilever beam. This is true because, numerically, the cantilever beam is actually the simply supported example assuming the symmetry conditions. Thus, presenting the cantilever beam was a strategic decision, because it encompasses both solutions.

Comment 5:

Finally, the author could give more information about the computational time of the performed simulation (as the whole duration for a single numerical convergence).

Answer: Following the reviewer suggestion, a new figure was included in the manuscript, in the end of section 3.1.1. This figure includes the computational cost of all analysis. Also a brief discussion of the computational cost was included in end of the same section 3.1.1. (red text) and in the “conclusions” section as well (blue text).

Comment 6:

Conclusion: In the conclusion section, the author could report the most significative results in quantitative terms (numbers and differences between the two numerical approaches). Moreover, further details on the future development or applications on the proposed RPIM method should be given.

Answer: Following the reviewer suggestion, the most significant results of the macroscale example were included in the “conclusions” section (red text). Also, in the same “conclusions” section it was inserted a text describing future developments and applications of RPIM with sandwich structures, using the data documented in this submitted manuscript (blue text)

Comment 7:

Language typos: Please correct some language typos (i.e. line 502 there is the repetition of “the”; line 596 change “bellow” in below”).

Answer: The typos were corrected, and the text was additionally revised.

Best regards,

Jorge Belinha